# FedEM: A Privacy-Preserving Framework for Concurrent Utility Preservation in Federated Learning

## Abstract

Federated Learning (FL) enables collaborative model training across distributed clients without sharing local data, thus reducing privacy risks in decentralized systems. However, the exposure of gradients during training can lead to significant privacy leakage, particularly under gradient inversion attacks. To address this issue, we propose Federated Error Minimization (FedEM), an input-level defense framework that injects learnable perturbations into client data and jointly optimizes both the model and the perturbation generator. Unlike traditional Differential Privacy methods that modify gradients, FedEM achieves a stricter privacy-utility trade-off by perturbing inputs directly. We validate the effectiveness of FedEM through extensive experiments on benchmark datasets. For example, on MNIST, FedEM achieves only a 0.08% decrease in accuracy compared to FedSGD, while significantly improving privacy metrics, with MSE improved by 46.2% and SSIM reduced by 69.3%. These results demonstrate that FedEM effectively mitigates gradient leakage attacks with minimal utility loss, providing a robust and scalable solution for privacy-preserving federated learning.

## 1 Introduction

Federated learning has emerged as a promising paradigm for collaborative machine learning, enabling multiple clients to jointly train a global model without directly sharing their local data (McMahan et al., 2017; Li et al., 2024). By preserving data decentralization, FL addresses privacy concerns while leveraging the diverse data distributions across clients. However, despite its advantages, FL is still vulnerable to privacy threats. Adversaries can exploit weaknesses in gradient-sharing techniques, which makes it challenging to design reliable and privacy-preserving FL systems.

Existing attack techniques, such as membership inference (Shokri et al., 2017), property inference (Melis et al., 2019), and gradient leakage attacks (GLAs) (Zhu et al., 2019), can compromise client privacy in FL environments. Among these, GLAs have drawn significant attention because they exploit shared gradients to recover the original training data, potentially revealing sensitive information about clients. These threats highlight the urgent need for effective privacy protection mechanisms in FL.

Several methods have been proposed to mitigate privacy risks in FL. Encryption-based techniques (Xu et al., 2019) offer robust privacy guarantees but introduce substantial computational and communication overhead, limiting scalability in resource-constrained environments. Differential privacy (DP) approaches, such as Centralized DP (CDP)(Geyer et al., 2017) and Local DP (LDP)(Sun et al., 2020), provide alternative solutions. However, these methods often degrade model performance due to the noise they introduce, particularly in LDP settings where noise is directly added to gradients. Achieving an optimal balance between privacy and utility remains a persistent challenge in FL research.

In this work, we draw inspiration from data poisoning techniques and introduce a novel algorithm, FedEM, aimed at enhancing privacy while minimizing performance degradation. Unlike traditional DP methods, which inject noise into gradients, FedEM incorporates controlled perturbations directly into the client data. These perturbations are carefully crafted to reduce the risk of data reconstruc-

tion while maintaining model utility. By reformulating the FL optimization objective to account for perturbation constraints, FedEM achieves a more favorable balance between utility and privacy protection. Through comprehensive experiments, we demonstrate that FedEM outperforms established privacy-preserving techniques in safeguarding privacy while maintaining competitive model performance.

The main contributions of this work are summarized as follows:

- We introduce FedEM, a novel data perturbation-based framework that innovatively defends against gradient leakage attacks while maintaining high model utility, offering a new perspective on achieving the privacy-utility trade-off.

- We design a dual-step optimization mechanism for FedEM: unlike traditional perturbation methods that fix perturbation intensity, FedEM integrates client-side multi-step perturbation update and parallel local model optimization, enabling adaptive adjustment of perturbation to balance utility and privacy.

- We thoroughly evaluate FedEM through extensive experiments on multiple benchmark datasets, demonstrating that it consistently outperforms prior state-of-the-art privacy-preserving techniques in achieving a superior privacy-utility trade-off.

## 2 RELATED WORKS

### 2.1 THREAT METHODS IN FEDERATED LEARNING

Federated learning enables multiple clients to collaboratively train machine learning models without sharing their raw data. While this decentralized paradigm offers a certain degree of privacy protection, it does not eliminate all privacy risks. Previous studies have shown that FL remains susceptible to information leakage through shared model updates. Several types of attacks have been investigated, including membership inference attacks (Shokri et al., 2017; Salem et al., 2018; Nasr et al., 2019) and property inference attacks (Ganju et al., 2018; Melis et al., 2019; Song & Mittal, 2021), which exploit the trained model to infer statistical or structural information about training data. However, with the development of defense strategies and a reassessment of their practical impact, recent research has shifted toward a more direct and severe threat: gradient leakage attacks, which aim to reconstruct raw training data from gradient information (Zhu et al., 2019; Zhao et al., 2020), posing a fundamental challenge to the privacy guarantees of federated learning.

Optimization-based gradient leakage attacks were first introduced by (Zhu et al., 2019), with the goal of reconstructing original training data (e.g., images) directly from shared gradients. Subsequent work, such as (Zhao et al., 2020), improved reconstruction quality, especially for high-resolution inputs. Later studies proposed regularization techniques to further enhance attack performance under large-scale or complex settings (Geiping et al., 2020; Yin et al., 2021; Yue et al., 2023). In addition to optimization-based methods, analytical approaches have been developed that leverage inherent properties of gradients to directly infer client data (Zhu & Blaschko, 2020; Chen & Campbell, 2021; Lu et al., 2022). Unlike iterative optimization, these techniques rely on explicit gradient analysis. Both optimization-based and analytical GLA methods generally assume a semi-honest server, which passively observes shared parameters without interfering in the FL process. However, recent studies have introduced malicious servers and clients (Fowl et al., 2021; 2022; Boenisch et al., 2023), who actively manipulate shared parameters or model weights, leading to more sophisticated attacks.

### 2.2 PRIVACY-PRESERVING MECHANISMS IN FEDERATED LEARNING

Privacy protection strategies in federated learning largely fall into two categories: encryption-based methods and DP-based techniques.

Encryption-based approaches, such as homomorphic encryption (HE) and secure multi-party computation (SMPC), aim to protect data during transmission and computation. HE enables computation on encrypted data without decryption (Aono et al., 2017; Madi et al., 2021), but its high computational and communication overhead limits scalability (Bonawitz et al., 2017). SMPC, which uses secret sharing to distribute data among multiple parties (Xu et al., 2019; Zhao et al., 2022), provides

strong privacy guarantees without requiring a trusted server. However, its considerable computational complexity hinders deployment in resource-constrained environments.

DP-based techniques are more commonly adopted in federated learning and are typically categorized into CDP and LDP (Jiang et al., 2024a). CDP methods (Geyer et al., 2017; Miao et al., 2022) assume a trusted server and inject noise during aggregation to mitigate membership and property inference attacks. While effective in those contexts, CDP offers limited protection against gradient leakage. In contrast, LDP adds noise directly to gradients before they are uploaded (Sun et al., 2020; Liu et al., 2020; Kim et al., 2021; Wang et al., 2023), offering stronger protection against gradient inversion. However, this noise often severely impairs model utility. To alleviate this, shuffling-based enhancements (Girgis et al., 2021) have been proposed, which reduce the required noise magnitude and improve the trade-off between privacy and performance. We summarize the most relevant works here and defer a more extensive survey to Appendix B.

## 3 THE FEDEM ALGORITHM FOR PRIVACY PROTECTION

### 3.1 FEDERATED LEARNING

We consider a federated learning system with $K$ clients, each holding a private dataset $\mathcal{D}_k$. The joint objective is to train a global model without sharing raw data:

$$\min_\theta \sum_{k=1}^{K} \frac{m_k}{m} \cdot \mathcal{L}_k(\theta),$$ (1)

where $m_k = |\mathcal{D}_k|$ and $m = \sum_{k=1}^{K} m_k$.

In each communication round, the server distributes the global model to clients, who then update it locally using their private data. The server subsequently aggregates these updates (e.g., FedAvg (McMahan et al., 2017)) to form a new global model. This iterative process continues until convergence and constitutes the standard FL pipeline, which serves as the basis for our FedEM framework.

### 3.2 THREAT MODEL

We assume all participants follow the prescribed federated training protocol. The server is modeled as honest-but-curious: it faithfully executes the protocol but may analyze received parameter updates to infer private client information. Consistent with standard assumptions, the server also knows the global model architecture and parameters.

For classification tasks, the ground-truth label $y$ can typically be inferred directly from the last-layer gradients (Zhao et al., 2020). Therefore, we assume $y$ is known to the server, and the attack focuses on recovering the input $x$. Formally, the attacker solves:

$$\min_x \ \left\| \nabla_\theta \mathcal{L}(x, y) - g \right\|,$$ (2)

where $\nabla_\theta \mathcal{L}(x, y)$ denotes the gradient with respect to model parameters computed on a candidate input $x$ with label $y$, and $g$ is the observed gradient from the client. By minimizing this discrepancy, the server can reconstruct inputs that closely approximate the original private data.

### 3.3 FEDEM

We propose a novel mechanism, FedEM, which introduces perturbations directly to clients' local data. By strategically injecting perturbations into the data, FedEM effectively defends against gradient leakage attacks while carefully controlling the magnitude of perturbations to minimize their impact on model performance.

With the introduction of data perturbation, let $\theta$ represent the global model parameters, and let $\delta_k$ denote the local perturbation vector for the $k$-th client, constrained by norm $\rho_u^{\min}$ and $\rho_u^{\max}$. The input features $x_k$ and corresponding labels $y_k$ are sampled from the local dataset $\mathcal{D}_k$, and the predictive model $f_\theta$ minimizes the loss function $\mathcal{L}$ applied to the perturbed data. The optimization objective in federated learning is reformulated as follows:

---

**Algorithm 1** FedEM (Federated Error-Minimization)

---

**Require:** Training datasets $\mathcal{D}_k$ (held by each client $k$); initial global model parameters $\theta$; local perturbation model parameters $\theta_u$; number of global rounds $T$; learning rate $\eta$; perturbation learning rate $\alpha_u$; number of perturbation steps $N$; perturbation norm bounds $\rho_u^{\min}, \rho_u^{\max}$

**Ensure:** Final model $\theta$

1: **Initialize:** $\theta$
2: **for** each round $t = 1$ to $T$ **do**
3:    Server selects a subset of clients $C_t$
4:    Server initializes perturbation $\delta_k$ for each $k \in C_t$ and sends $\theta$ to clients
5:    Each client traverses its full local dataset $\mathcal{D}_k$ in batches
6:    **for** each batch index (shared across $C_t$) **do**
7:      **for** each client $k \in C_t$ **in parallel do**
8:        Sample batch $(x_k, y_k) \sim \mathcal{D}_k$, set $\theta_u \leftarrow \theta$
9:        **for** step $n = 1$ to $N$ **do**
10:          $\delta_k \leftarrow \delta_k - \alpha_u \cdot \text{sign}(\nabla_{\delta_k} \mathcal{L}_k(f_{\theta_u}(x_k + \delta_k), y_k))$
11:          Project $\delta_k$ to norm constraint: $\delta_k \leftarrow \text{Proj}_{\rho_u^{\min} \leq \|\delta_k\| \leq \rho_u^{\max}}(\delta_k)$
12:          $\theta_u \leftarrow \theta_u - \eta \cdot \nabla_{\theta_u} \mathcal{L}_k(f_{\theta_u}(x_k + \delta_k), y_k)$
13:        **end for**
14:        Upload $g_k = \nabla_\theta \mathcal{L}_k(f_\theta(x_k + \delta_k), y_k)$
15:      **end for**
16:      $\theta \leftarrow \theta - \eta \cdot \frac{1}{|C_t|} \sum_{k \in C_t} g_k$
17:    **end for**
18: **end for**
19: **return** Trained global model parameters $\theta$

---

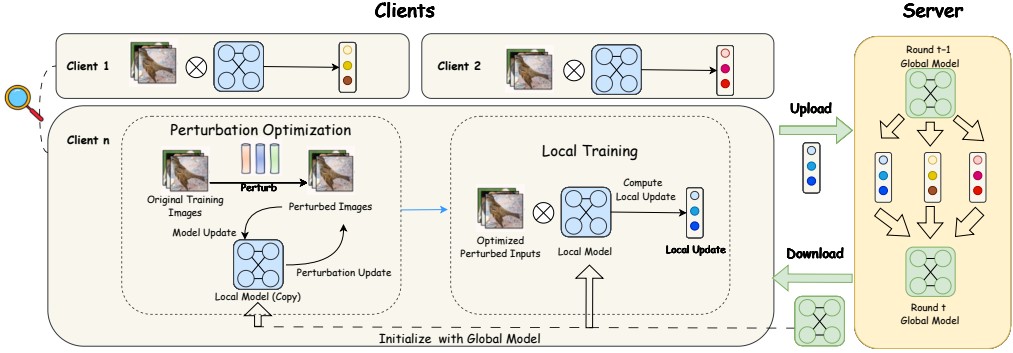

Figure 1: Overview of the FedEM framework. At the beginning of each round, the server distributes the global model to selected clients. Each client performs *perturbation optimization*, where both the local perturbation and local model are updated iteratively. The resulting perturbed inputs are then used in *local training* to compute gradient updates, which are uploaded to the server. The server aggregates all client updates to refresh the global model.

$$\min_\theta \min_{\delta_1, \delta_2, \ldots, \delta_k} \quad \sum_{k=1}^{K} \frac{m_k}{m} \mathbb{E}_{(x_k, y_k) \sim \mathcal{D}_k} \left[ \mathcal{L}\left(f_\theta\left(x_k + \delta_k\right), y_k\right) \right] \tag{3}$$

$$\text{s.t.,} \quad \rho_u^{\min} \leq \|\delta_k\| \leq \rho_u^{\max}.$$

To solve the above optimization problem, FedEM employs an iterative client-server federated training framework with integrated input-space perturbation. At the beginning of each global communication round, the server selects a subset of clients $C_t$ and broadcasts the current global model parameters $\theta$ along with an initial perturbation vector $\delta_k$ for each selected client $k \in C_t$. Each client then partitions its local dataset $\mathcal{D}_k$ into mini-batches and sequentially traverses all batches. For each

batch $(x_k, y_k)$, the client initializes a local perturbation model copy $\theta_u \leftarrow \theta$. Over $N$ inner steps, the client updates the perturbation vector $\delta_k$ using projected gradient descent on the loss with respect to $\delta_k$:

$$\delta_k \leftarrow \text{Proj}_{\rho_u^{\min} \leq ||\delta_k|| \leq \rho_u^{\max}} \left( \delta_k - \alpha_u \cdot \text{sign} \left( \nabla_{\delta_k} \mathcal{L}_k (f_{\theta_u}(x_k + \delta_k), y_k) \right) \right), \tag{4}$$

ensuring that the perturbation remains within a bounded $L_2$ norm ball. In parallel, the local perturbation model $\theta_u$ is updated via gradient descent:

$$\theta_u \leftarrow \theta_u - \eta \cdot \nabla_{\theta_u} \mathcal{L}_k (f_{\theta_u}(x_k + \delta_k), y_k). \tag{5}$$

After completing $N$ perturbation steps for the current batch, the client computes the gradient of the original global model $\theta$ using the perturbed input: $g_k = \nabla_\theta \mathcal{L}_k (f_\theta(x_k + \delta_k), y_k)$, and uploads $g_k$ to the server. The server aggregates the gradients received from all selected clients for this batch, averages them, and immediately performs a model update:

$$\theta \leftarrow \theta - \eta \cdot g_{\text{global}}, \tag{6}$$

where $g_{\text{global}} = \frac{1}{|C_t|} \sum_{k \in C_t} g_k$. This procedure repeats over all local batches and across $T$ global communication rounds. The complete algorithm is provided in Algorithm 1, and its structural overview is illustrated in Figure 1. A complete description of all the notations used throughout the paper is provided in Appendix A.

### 3.4 CONVERGENCE ANALYSIS

We provide a theoretical guarantee for FedEM under standard smoothness and bounded variance assumptions. The complete assumptions, lemmas, and detailed proofs are deferred to Appendix E.

**Theorem 1** (Convergence of FedEM). *Let $f(\theta) = \sum_{k=1}^{K} \frac{m_k}{m} f_k(\theta)$ be the global objective, assume $f$ is $L$-smooth and stochastic gradients have bounded variance. Suppose each client perturbation $\delta_k$ is bounded by $\|\delta_k\| \leq \rho_u^{max}$ and client heterogeneity is bounded by $\zeta^2$. Then with step size $\eta \leq \frac{1}{6L}$, after $T$ updates FedEM satisfies*

$$\frac{1}{T} \sum_{t=0}^{T-1} \mathbb{E}\left[\|\nabla f(\theta^t)\|^2\right] = \mathcal{O}\left(\frac{1}{\sqrt{T}}\right) + \mathcal{O}(\rho_u^{max2}) + \mathcal{O}(\zeta^2).$$

Theorem 1 shows that FedEM converges to a neighborhood of stationary points, with the neighborhood size controlled by the perturbation radius $\rho_u^{\max}$. Smaller perturbations tighten convergence but offer weaker privacy, while larger perturbations enhance privacy at the cost of model accuracy.

## 4 EXPERIMENTS

### 4.1 EXPERIMENTAL SETUPS

**Datasets, Baselines, and Evaluation Metrics.** We conduct experiments on three widely used benchmark datasets in federated learning: MNIST (LeCun et al., 1998), FashionMNIST (Xiao et al., 2017), CIFAR-10, CIFAR-100 (Krizhevsky, 2009) and Tiny-imagenet, to evaluate the effectiveness of the proposed FedEM algorithm. For comparison, we select a variety of privacy-preserving methods as baselines, including standard local differential privacy (LDP) mechanisms Wei et al. (2021) with both Gaussian and Laplace noise, PPFA (Zhang et al., 2023), and LDPM (Jiang et al., 2024b). We evaluate model utility using validation and test accuracy. To assess privacy protection, we measure the quality of reconstructed images obtained by attackers using metrics such as Mean Squared Error (MSE), Structural Similarity Index Measure (SSIM)(Wang et al., 2004), Peak Signal-to-Noise Ratio (PSNR), Learned Perceptual Image Patch Similarity (LPIPS)(Zhang et al., 2018), and Kullback-Leibler (KL) divergence, which together quantify the difference between reconstructed and original samples.

Table 1: Main experimental results across five datasets (MNIST, FMNIST, CIFAR-10, CIFAR-100, and Tiny-ImageNet). Utility metrics are marked with **U**, and privacy metrics with **P**. Arrows indicate preferred direction: ↑ = higher is better, ↓ = lower is better.

| Dataset | Method | Val Acc (U↑) | Test Acc (U↑) | MSE (P↑) | PSNR (P↓) | SSIM (P↓) | LPIPS (P↑) | KL (P↑) |
|---|---|---|---|---|---|---|---|---|
| MNIST | DP-Gas | 0.9774 | 0.9741 | 1.3721 | 9.2340 | 0.1013 | 0.6321 | 3.3368 |
| | DP-Lap | 0.9733 | 0.9717 | 1.3970 | 8.8633 | 0.0455 | 0.6529 | 2.9662 |
| | PPFA | 0.9663 | 0.9573 | 1.2509 | 9.3820 | 0.0932 | 0.6109 | 3.5123 |
| | LDPM | 0.9756 | 0.9749 | 1.6201 | 8.2451 | 0.0527 | 0.6444 | 3.8519 |
| | **FedEM (ours)** | **0.9809** | **0.9767** | **1.8251** | **7.6982** | **0.0378** | **0.6715** | **4.5235** |
| FMNIST | DP-Gas | 0.8664 | 0.8543 | 1.1693 | 8.6704 | 0.1910 | 0.5806 | 2.5319 |
| | DP-Lap | 0.8665 | 0.8497 | 1.3012 | 8.3033 | 0.1158 | 0.6052 | 1.6939 |
| | PPFA | 0.8473 | 0.8375 | 1.3615 | 8.1524 | 0.1297 | 0.5892 | 2.1581 |
| | LDPM | 0.8715 | 0.8527 | 1.4241 | 7.8580 | 0.0877 | 0.5809 | 2.3729 |
| | **FedEM (ours)** | **0.8719** | **0.8592** | **1.4988** | **7.4209** | **0.0501** | **0.6140** | **2.8601** |
| CIFAR-10 | DP-Gas | 0.2449 | 0.2504 | 1.8638 | 9.4538 | 0.0144 | 0.7549 | 2.6632 |
| | DP-Lap | 0.2195 | 0.2213 | 1.9974 | 9.2169 | 0.0153 | 0.7601 | 2.7273 |
| | PPFA | 0.2489 | 0.2505 | 1.8693 | 9.4146 | 0.0152 | 0.7548 | 2.0903 |
| | LDPM | 0.2277 | 0.2278 | 2.0565 | 9.0540 | 0.0170 | 0.7455 | 3.2811 |
| | **FedEM (ours)** | **0.2502** | **0.2518** | **2.0685** | **9.0501** | **0.0140** | **0.7954** | **3.3572** |
| CIFAR-100 | DP-Gas | 0.2911 | 0.2839 | 2.2745 | 8.0503 | 0.0344 | 0.6811 | 2.4578 |
| | DP-Lap | 0.2857 | 0.2865 | 1.9363 | 8.7527 | **0.0273** | 0.6644 | 3.1130 |
| | PPFA | 0.2815 | 0.2753 | 2.1107 | 8.2862 | 0.0421 | 0.6813 | 3.2916 |
| | LDPM | 0.2833 | 0.2753 | 2.2968 | 8.0068 | 0.0427 | 0.7072 | 2.9708 |
| | **FedEM (ours)** | **0.2947** | **0.2870** | **2.3854** | **7.9706** | 0.0303 | **0.7321** | **3.5712** |
| Tiny-ImageNet | DP-Gas | 0.1495 | 0.1519 | 1.9134 | 8.6360 | 0.0361 | **0.7813** | 6.1659 |
| | DP-Lap | 0.1563 | 0.1587 | 1.9253 | 8.4487 | 0.0130 | 0.7317 | 4.6338 |
| | PPFA | 0.1525 | 0.1574 | 1.9025 | 8.8802 | 0.0150 | 0.7411 | 6.1615 |
| | LDPM | 0.1603 | 0.1618 | 1.9268 | 8.4821 | 0.0132 | 0.7746 | 5.6384 |
| | **FedEM (ours)** | **0.1612** | **0.1633** | **1.9336** | **8.3714** | **0.0120** | 0.7726 | **6.2263** |

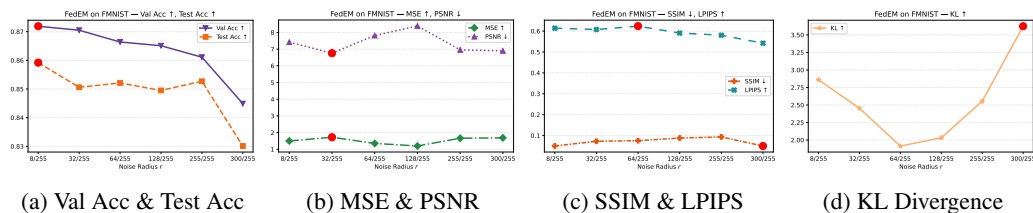

| (a) Val Acc & Test Acc | (b) MSE & PSNR | (c) SSIM & LPIPS | (d) KL Divergence |
|---|---|---|---|

Figure 2: FedEM on FMNIST: Metric trends under varying $L_2$-norm radii $r$ in E1. Red dots indicate best-performing radii for each metric.

**FL Settings.** By default, the federated learning system consists of 4 clients. The global training process runs for 30 communication rounds, with each client performing 1 local training epoch per round. The default local batch size is set to 8. All datasets are split into 70% training, 15% validation, and 15% testing, with data equally partitioned among clients. For MNIST and FashionMNIST, we adopt the LeNet architecture, and for CIFAR-10 we use the ConvNet-64 model. Both local model updates and perturbation generation are optimized using SGD with a learning rate of 0.1 and no weight decay. We adopt the Invert-Grad method (Geiping et al., 2020)—one of

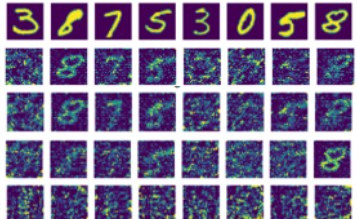

Figure 3: Reconstructed MNIST samples (top to bottom): Original, FedSGD, PPFA, DP-Gas, FedEM.

the most widely used and representative gradient inversion attack paradigms in existing literature, as the attack model. By default, the perturbation is generated using PGD under $L_2$ norm. Further implementation details can be found in Appendix D.1.

## 4.2 Main Results

To ensure a fair comparison under high-utility settings, we set the privacy budgets or noise scales of each baseline as follows: for LDP methods, the noise scales is fixed at $1/255$; for PPFA, we set $\epsilon$=0.995; for LDPM, we use a noise scale of $\sigma$=0.0005; and for FedEM, the perturbation radius is set to $8/255$. For the utility metrics, we report both the validation and test accuracy as the final performance indicators after the model has converged. For the privacy metrics, we select the results from the first global training round (E1) when the gradient leakage attack is launched.

As summarized in Table 1, FedEM consistently achieves state-of-the-art performance across five datasets with varying complexity, ranging from simple handwritten digits (MNIST) to more chal-

lenging large-scale benchmarks (CIFAR-100 and Tiny-ImageNet). On MNIST and FashionMNIST, FedEM yields the highest validation and test accuracy while offering the strongest resistance against gradient leakage. Figure 3 shows the reconstructed MNIST samples. On CIFAR-10, although all methods exhibit degraded performance due to the dataset's complexity, FedEM still maintains the best trade-off. Notably, on CIFAR-100 and Tiny-ImageNet, which present significantly more challenging and diverse distributions, FedEM preserves its advantage, achieving both superior utility and stronger privacy protection compared to existing defenses. These results highlight FedEM's robustness and scalability, demonstrating that it generalizes effectively across heterogeneous data domains and remains effective even under large-scale, high-dimensional federated learning tasks.

To further illustrate the evolution of the perturbations, we visualize them during FedEM training using a CIFAR-10 image as an example. As shown in Fig. 4: initially, they are nearly imperceptible random noise, but as training progresses, they evolve into structured patterns, highlighting the dynamic role of defensive noise in the learning process.

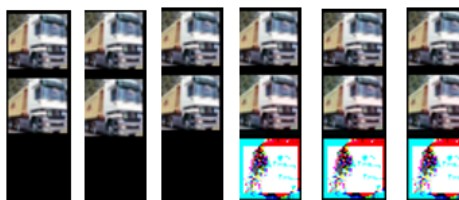

Figure 4: Evolution of perturbations in FedEM at different perturbation steps (1, 5, 10, 15, 30, 50). Top: original image; middle: perturbed image; bottom: normalized perturbation map. Perturbations are rescaled for visibility, but remain imperceptible to the human eye in the perturbed images.

### 4.3 EXTENSION TO TEXT DATA UNDER GRADIENT LEAKAGE ATTACKS

To assess the generalizability of FedEM beyond image-based tasks, we conduct experiments on the CoLA dataset for text classification. We simulate federated training with a batch size of 1 and 10 communication rounds per client, and apply the LAMP gradient inversion attack (Balunovic et al., 2022). Utility is measured using Matthews Correlation Coefficient (MCC), while privacy leakage is quantified by ROUGE scores (Lin, 2004) on the reconstructed text.

To adapt FedEM to language models, we inject $L_2$-bounded perturbations into the embedding space with a radius of 2.0. As shown in Table 2, FedEM exhibits minimal token-level leakage in qualitative results, with least recognizable tokens reconstructed—unlike other baselines. This demonstrates that our perturbation strategy can be successfully extended from continuous input spaces (e.g., images) to discrete input representations (e.g., word

Table 2: Reconstructed sentences under gradient leakage on CoLA. Tokens matching the original input are highlighted to reflect privacy leakage.

| Original | john tries to meet not mary. |
|---|---|
| **FedSGD** | john tries not to meet mary. |
| **DP-SGD** | john tries not meet maryumatic |
| **Grad-Masked** | alyssa not mary tries meet john. |
| **FedEM (ours)** | .tries to undergoneanalysis. |

embeddings). Corresponding quantitative results are reported in Table 3. While FedSGD achieves slightly higher MCC due to its lack of defense, it suffers severe leakage across all ROUGE metrics. In contrast, FedEM achieves the lowest ROUGE-1/2/L scores, indicating significantly reduced reconstruction quality, while maintaining competitive utility. These results confirm that FedEM effectively limits gradient-based text recovery attacks in discrete domains without sacrificing task performance. Detailed experimental settings and additional results on other text datasets can be found in Appendix D.4.

Table 3: Performance on the CoLA dataset under gradient leakage attack. MCC indicates utility (↑), while ROUGE-1/2/L (%) measure reconstruction quality of leaked text (↓). Utility metrics are marked with **U**, and privacy metrics with **P**

| Method | MCC (U↑) | ROUGE-1 (P↓) | ROUGE-2 (P↓) | ROUGE-L (P↓) |
|---|---|---|---|---|
| FedSGD (no defend) | **0.557** | 88.3 | 59.6 | 81.2 |
| DP-SGD | 0.551 | 81.2 | 42.7 | 69.4 |
| Gradient Masked | 0.555 | 83.7 | 53.2 | 76.7 |
| FedEM (ours) | 0.553 | **79.6** | **26.1** | **63.1** |

Table 4: Performance comparison with different methods under large-scale scenarios (50 clients). Utility metrics are marked with **U**, and privacy metrics with **P**. Arrows indicate preferred direction: ↑ = higher is better, ↓ = lower is better.

| Dataset | Method | Val Acc (U↑) | Test Acc (U↑) | MSE (P↑) | PSNR (P↓) | SSIM (P↓) | LPIPS (P↑) | KL (P↑) |
|---|---|---|---|---|---|---|---|---|
| MNIST | DP-Gas | 0.9671 | 0.9671 | 4.5111 | 3.7008 | 0.0192 | 0.7549 | 4.2343 |
| | DP-Lap | 0.9619 | 0.9622 | 4.7501 | 3.4704 | **0.0138** | 0.7682 | 4.2947 |
| | PPFA | 0.9651 | 0.9661 | 4.5924 | 3.6321 | 0.0171 | 0.7544 | 4.3046 |
| | LDPM | 0.9650 | 0.9647 | 3.1847 | 5.3353 | 0.1081 | 0.5794 | 4.2762 |
| | **FedEM (ours)** | **0.9691** | **0.9689** | 4.8032 | 3.4150 | 0.0168 | **0.7685** | **4.7410** |
| FMNIST | DP-Gas | 0.8899 | 0.8894 | 0.6693 | 10.990 | 0.3649 | 0.3758 | 1.8011 |
| | DP-Lap | 0.8880 | 0.8889 | 0.7691 | 10.320 | 0.2553 | **0.4664** | 2.8238 |
| | PPFA | 0.8908 | 0.8909 | 0.7088 | 10.908 | 0.2748 | 0.4322 | **3.7652** |
| | LDPM | 0.8882 | 0.8885 | 0.7700 | 10.309 | 0.2549 | 0.3926 | 2.2443 |
| | **FedEM (ours)** | **0.8920** | **0.8911** | 0.8343 | 10.034 | 0.2455 | 0.4224 | 2.8061 |
| CIFAR-10 | DP-Gas | 0.4413 | 0.4420 | 1.9123 | 9.4159 | 0.0323 | 0.7210 | 2.8953 |
| | DP-Lap | 0.4800 | 0.4791 | 1.8817 | 9.5513 | 0.0316 | 0.7181 | **3.1515** |
| | PPFA | 0.4897 | 0.4918 | 1.7081 | 9.9190 | 0.0245 | 0.7167 | 2.0933 |
| | LDPM | 0.4933 | 0.4962 | 1.4511 | 10.487 | 0.0293 | 0.7102 | 2.5379 |
| | **FedEM (ours)** | **0.5267** | **0.5238** | 1.9436 | 9.2037 | **0.0235** | **0.7265** | 2.6642 |

## 4.4 Scalability under Large Client Participation

To further evaluate the scalability of our approach, we extend the experiments to a large-scale scenario involving 50 clients. The results on MNIST, FMNIST, and CIFAR-10 are reported in Table 4. FedEM consistently achieves the strongest overall performance, maintaining both high accuracy and robust privacy protection. On MNIST and FMNIST, it provides marginal gains in accuracy over baselines, while delivering superior privacy robustness, reflected in higher MSE, lower PSNR, and competitive LPIPS/KL scores. On CIFAR-10, which poses greater challenges due to high client heterogeneity, FedEM achieves a substantial accuracy improvement (exceeding the best baseline by over 3%) while simultaneously preserving stronger privacy guarantees. These results demonstrate that FedEM scales effectively to settings with large client participation, confirming its robustness under more realistic federated learning conditions.

## 4.5 Effect of Perturbation Magnitude on Privacy-Utility Trade-off

To mitigate the influence of randomness and evaluate the robustness of our approach, we further investigate the performance of different privacy-preserving algorithms under varying perturbation magnitudes. Using the same evaluation metrics introduced in Section 4.2, we plot line charts for each metric. Metrics with similar functionality or value range are grouped within the same subplot. The metric trends for FedEM under different $L_2$-norm radii on the FMNIST, CIFAR-10 and MNIST datasets are shown in Figures 2, 5 and 9 (see Appendix D.5.1), respectively.

Overall, across all datasets, we observe a consistent pattern: utility performance (e.g., test accuracy) generally declines as the perturbation strength increases. However, the relationship between privacy strength and noise magnitude is not strictly monotonic. In particular, for our proposed method FedEM, a moderate increase in perturbation radius initially leads to stronger privacy protection—as evidenced by improvements in privacy metrics such as LPIPS and MSE—but excessive noise often results in diminishing or fluctuating privacy gains. In contrast, baseline methods (see Appendix D.5.2) such as GasDP and PPFA exhibit a more straightforward pattern: stronger perturbation yields better privacy at the cost of rapidly degraded utility. Remarkably, FedEM achieves comparable or even stronger privacy protection at lower noise levels. This highlights that FedEM strikes a more favorable privacy-utility trade-off, and indicates the advantage of learning-based perturbation mechanisms in flexibly balancing objectives. Comprehensive experimental results for all noise scales and datasets are deferred to Appendix D.5.

## 4.6 Generalization of FedEM to Stronger Gradient Leakage Attacks

Table 5: Evaluation of FedEM under the GIAS(Yin et al., 2021) gradient-leakage attack on CIFAR-100.

| Attack | Method | Val Acc (U↑) | Test Acc (U↑) | MSE (P↑) | PSNR (P↓) | SSIM (P↓) | LPIPS (P↑) | KL (P↑) |
|---|---|---|---|---|---|---|---|---|
| GIAS | DP-Gas | 0.2911 | 0.2839 | 1.5864 | 10.123 | 0.0387 | 0.6237 | 3.0508 |
| | DP-Lap | 0.2857 | 0.2865 | 1.7448 | 9.6432 | 0.0356 | 0.6660 | **3.3219** |
| | PPFA | 0.2815 | 0.2753 | 1.7318 | 9.9885 | 0.0348 | 0.6552 | 3.1252 |
| | LDPM | 0.2833 | 0.2753 | 1.5480 | 10.192 | 0.0318 | 0.6456 | 3.2615 |
| | **FedEM (ours)** | **0.2947** | **0.2870** | 1.7513 | 9.4589 | **0.0286** | **0.6729** | 3.2531 |

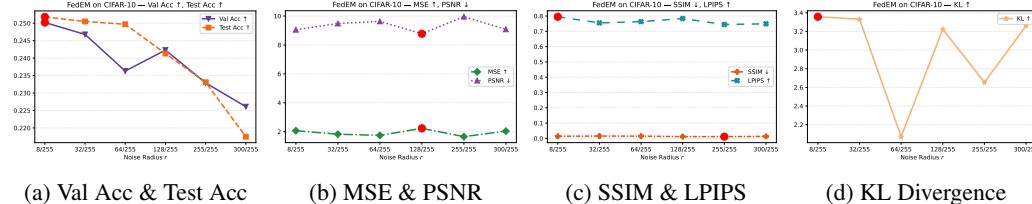

| (a) Val Acc & Test Acc | (b) MSE & PSNR | (c) SSIM & LPIPS | (d) KL Divergence |

Figure 5: FedEM on CIFAR-10: E1-round performance on all metrics across different $L_2$ radii. Best performance points are highlighted.

To further evaluate the robustness of FedEM beyond the Inverting-Grad attack, we test its performance under the GIAS attack (Yin et al., 2021) on CIFAR-100. The results are summarized in Table 5. Compared with baselines, FedEM attains the best performance on most metrics. The utility metrics remain stable under different attacks. On the privacy side, FedEM yields the largest MSE and the highest LPIPS, and it achieves the lowest PSNR among the compared methods. While DP-Lap achieves a marginally higher KL divergence, FedEM provides a more consistent advantage across the suite of privacy metrics. These results demonstrate that FedEM generalizes to other reconstruction-based attacks and further validate its robustness and applicability under diverse federated-learning threat models.

### 4.7 IMPACT OF PERTURBATION LOWER BOUNDS ON FEDEM PERFORMANCE

To further justify the design of FedEM and its use of input perturbation constraints, we experimentally verify a key theoretical insight (lemma 2) proposed in (Zhang et al., 2024): when the applied perturbation has a non-zero lower bound, the resulting privacy leakage remains upper bounded (see Appendix C for detailed discussion). In this study, we vary the lower bound $\rho_u^{\min}$ of the perturbation norm while keeping the upper bound $\rho_u^{\max}$ fixed, and apply gradient leakage attacks in the first training round. For a detailed comparison of FedEM under different lower bound settings on CIFAR-10 (E1), see Table 6. FedSGD, which applies no perturbation, serves as the baseline. Results show that even small non-zero $\rho_u^{\min}$ values already lead to substantial privacy improvements over FedSGD. Increasing $\rho_u^{\min}$ further does not consistently yield better privacy, suggesting diminishing returns. Importantly, across all settings with non-zero perturbation, the privacy leakage remains bounded—confirming Lemma 2, which states that once the distortion exceeds a certain threshold, the privacy loss is upper bounded regardless of the exact lower bound. Comprehensive experimental results are provided in Appendix D.6. (Due to space limitations, we provide convergence analysis and error robustness experiments in Appendix D.2 and D.3.)

Table 6: FedEM performance under different perturbation lower bounds $\rho_u^{\min}$ (with fixed upper bound $\rho_u^{\max}$) on CIFAR-10, evaluated at training epoch E1. Colors are used to show performance differences relative to the baseline: (light green) indicates increase in performance, and (light orange) indicates decrease in performance.

| Method | Val Acc (U↑) | Test Acc (U↑) | MSE (P↑) | PSNR (P↓) | SSIM (P↓) | LPIPS (P↑) | KL (P↑) |
|---|---|---|---|---|---|---|---|
| FedSGD (baseline) | 0.2803 | 0.2771 | 1.8219 | 9.5554 | 0.0178 | 0.7556 | 2.9228 |
| $\rho_u^{\max} = 200/255, \rho_u^{\min} = 25/255$ | 0.2419 (-0.0384) | 0.2479 (-0.0292) | 1.8512 (+0.0293) | 9.4701 (-0.0857) | 0.0157 (-0.0021) | 0.7564 (+0.0008) | 3.0267 (+0.1039) |
| $\rho_u^{\max} = 200/255, \rho_u^{\min} = 50/255$ | 0.2377 (-0.0426) | 0.2375 (-0.0396) | 1.9897 (+0.1678) | 9.1817 (-0.3737) | 0.0149 (-0.0029) | 0.7614 (+0.0058) | 3.6679 (+0.7451) |
| $\rho_u^{\max} = 200/255, \rho_u^{\min} = 100/255$ | 0.2061 (-0.0742) | 0.2225 (-0.0546) | 2.0333 (+0.2114) | 9.1708 (-0.3846) | 0.0147 (-0.0031) | 0.7686 (+0.0130) | 2.9418 (+0.0190) |
| $\rho_u^{\max} = 400/255, \rho_u^{\min} = 50/255$ | 0.2167 (-0.0636) | 0.2283 (-0.0488) | 1.9006 (+0.0787) | 9.2962 (-0.2592) | 0.0147 (-0.0031) | 0.7615 (+0.0059) | 3.1473 (+0.2245) |
| $\rho_u^{\max} = 400/255, \rho_u^{\min} = 100/255$ | 0.1935 (-0.0868) | 0.1957 (-0.0814) | 1.9620 (+0.1401) | 9.1756 (-0.3798) | 0.0140 (-0.0038) | 0.7630 (+0.0074) | 3.0510 (+0.1282) |
| $\rho_u^{\max} = 400/255, \rho_u^{\min} = 200/255$ | 0.1827 (-0.0976) | 0.2029 (-0.0742) | 1.9163 (+0.0944) | 9.2915 (-0.2639) | 0.0132 (-0.0046) | 0.7591 (+0.0035) | 3.7197 (+0.7970) |

## 5 CONCLUSION

In this paper, we propose FedEM, a data perturbation-based federated learning framework designed to defend against gradient leakage attacks. Unlike most existing defenses that operate on gradients, FedEM directly perturbs client inputs to preserve model utility while reducing the risk of inversion-based privacy leakage, and comprehensive evaluations across image and text tasks demonstrate that FedEM achieves a more favorable privacy-utility trade-off compared to prior methods. We believe the proposed perturbation-based formulation opens up new possibilities for scalable, privacy-aware learning, and we encourage future work to explore its applicability to other tasks such as robustness enhancement, fairness enforcement, and personalized federated learning.

## ETHICS STATEMENT

This work adheres to the ICLR Code of Ethics. All datasets used were sourced in compliance with relevant usage guidelines, ensuring no violation of privacy. No personally identifiable information was used, and no experiments were conducted that could raise privacy or security concerns.

## REPRODUCIBILITY STATEMENT

We have made every effort to ensure that the results presented in this paper are reproducible. All code and datasets are provided in the supplementary material to facilitate replication and verification. The experimental setup, including training steps, model configurations, and hardware details, is described in detail in the paper.

## LLM USAGE

Large Language Models (LLMs) were used to aid in the writing and polishing of the manuscript. Specifically, we used an LLM to assist in refining the language, improving readability, and ensuring clarity in various sections of the paper. The model helped with tasks such as sentence rephrasing, grammar checking, and enhancing the overall flow of the text. It is important to note that the LLM was not involved in the ideation, research methodology, or experimental design. All research concepts, ideas, and analyses were developed and conducted by the authors. The contributions of the LLM were solely focused on improving the linguistic quality of the paper, with no involvement in the scientific content or data analysis.

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

## A    NOTATION SUMMARY

| Symbol | Description |
|--------|-------------|
| $K$ | Number of clients in federated learning |
| $\mathcal{D}_k$ | Local dataset of client $k$ |
| $m_k$ | Number of data points on client $k$, $m_k = |\mathcal{D}_k|$ |
| $m$ | Total number of data points, $m = \sum_k m_k$ |
| $\theta$ | Global model parameters |
| $\theta_u$ | Local perturbation model used to update $\delta_k$ |
| $C_t$ | Set of selected clients in global round $t$ |
| $(x_k, y_k)$ | Input features and labels from client $k$ |
| $f_\theta(\cdot)$ | Prediction model parameterized by $\theta$ |
| $\mathcal{L}_k(\cdot)$ | Loss function of client $k$ |
| $g_k$ | Gradient from client $k$: $g_k = \nabla_\theta \mathcal{L}_k(f_\theta(x_k + \delta_k), y_k)$ |
| $g_{\text{global}}$ | Aggregated gradient across clients |
| $\delta_k$ | Perturbation vector added to client $k$'s input |
| $\rho_u^{\min}, \rho_u^{\max}$ | Lower and upper bounds on $\|\delta_k\|$ |
| $\alpha_u$ | Learning rate for perturbation updates |
| $N$ | Number of local perturbation steps per batch |
| $\epsilon_p$ | Privacy leakage score (reconstruction-based) |
| $x^{(m)}$ | Ground truth data sample |
| $x_i^{(m)}$ | Reconstructed sample at attack iteration $i$ |
| $D$ | Maximum possible reconstruction distance |
| $I$ | Number of attacker optimization iterations |
| $\Delta$ | Mean distortion between original and perturbed data |

Table 7: Summary of notations used throughout the paper.

## B    RELATED WORKS

### B.1    ADVERSARIAL TRAINING

Adversarial training has emerged as a canonical defense mechanism against adversarial perturbations, aiming to reinforce the resilience of deep neural networks when confronted with deliberately manipulated inputs. Rather than relying solely on clean data, the model is exposed during training to inputs that are perturbed within a constrained set, thereby encouraging it to learn decision boundaries that are less sensitive to small but malicious changes. This idea can be formalized as a minimax optimization problem in which the learner minimizes the predictive loss while simultaneously considering the worst-case perturbation under a bounded norm. Specifically, for a classifier $f$

parameterized by $\theta$, the objective is expressed as

$$\min_{\theta} \max_{\delta} \; \mathbb{E}_{(x,y)\sim\mathcal{D}}[\mathcal{L}(f(x+\delta;\theta),y)], \quad \text{s.t., } ||\delta|| \leq \epsilon. \tag{7}$$

where $\delta$ denotes the perturbation constrained by a $p$-norm budget $\varepsilon$, $x + \delta$ represents the adversarial input, and $\mathcal{L}$ is the loss relative to the true label $y$.

In practice, this training regime alternates between two phases. The inner maximization step generates perturbed samples that induce the largest possible loss within the allowable perturbation set, often constructed using gradient-based techniques such as PGD. The outer minimization step then updates the model parameters by minimizing the empirical risk on these perturbed samples.

## B.2  Error Minimization Attack.

The error minimization attack (EMA), introduced by Zheng et al. (Zheng et al., 2020), aims to poison the training process by embedding subtle perturbations into training inputs. Unlike traditional adversarial methods that maximize the model's loss to impair learning, EMA adopts a min-min formulation, where both model parameters and perturbations are optimized to minimize the loss:

$$\min_{\theta} \min_{\delta} \; \mathbb{E}_{(x,y)\sim\mathcal{D}}[\mathcal{L}(f(x+\delta;\theta),y)], \quad \text{s.t., } ||\delta|| \leq \epsilon. \tag{8}$$

This approach preserves model utility during training while introducing hard-to-detect biases into the learned representations. In contrast to unlearnable examples (Huang et al., 2021), which use a min-max structure to prevent unauthorized learning by degrading performance, EMA maintains high accuracy but compromises the integrity of training. In our work, we draw inspiration from EMA and reinterpret its optimization structure as a privacy defense strategy: carefully designed perturbations are leveraged to mitigate gradient leakage without harming utility.

# C  Theoretical Discussion

Although our work does not propose new theoretical results, we include in this appendix two key lemmas from (Zhang et al., 2024) that help support the design rationale behind our algorithm. These results establish a theoretical relationship between the extent of data distortion and the upper bound of privacy leakage in federated learning.

Specifically, we revisit the formal privacy metric defined in Eq. 9, and present two lemmas that show how adversarial reconstruction capabilities are limited when sufficient perturbation is applied. We reproduce their derivations here for completeness and to provide theoretical intuition for the distortion constraints used in FedEM.

## C.1  Measurement for Data Privacy

We adopt the definition of privacy leakage proposed in (Zhang et al., 2024), which quantifies the amount of private information that can be inferred by an adversary during model inversion. Let $x^{(m)}$ denote the original $m$-th data sample, and $x_i^{(m)}$ the reconstruction of this sample inferred by the attacker at iteration $i$. Let $D$ be a positive constant representing the maximum possible distance between original and reconstructed samples. The total number of attack iterations is denoted by $I$. The privacy leakage $\epsilon_p$ is then defined as:

$$\epsilon_p = \begin{cases} \dfrac{D - \frac{1}{I}\sum\limits_{i=1}^{I}\frac{1}{|\mathcal{D}|}\sum\limits_{m=1}^{|\mathcal{D}|}\|x_i^{(m)}-x^{(m)}\|}{D}, & I > 0 \\ 0, & I = 0 \end{cases} \tag{9}$$

This normalized score reflects the average reconstruction accuracy achieved by the attacker: higher values of $\epsilon_p$ correspond to more successful inference and therefore more severe privacy leakage.

## C.2 THEORETICAL CONNECTION BETWEEN DISTORTION AND PRIVACY LEAKAGE

Building upon the privacy metric defined in Eq. 9, we now establish theoretical guarantees that connect the degree of data distortion with the upper bound on privacy leakage. The following lemma (Zhang et al., 2024) provides an upper threshold on $\epsilon_p$ as a function of the distortion extent and the attacker's optimization capability.

**Lemma 1** (Upper Bound on Privacy Leakage (Zhang et al., 2024)). *Consider a semi-honest adversary that reconstructs client data through an optimization-based inversion attack. Let $\Delta$ denote the distortion extent between the original and perturbed data, defined as $\Delta = \left\| \frac{1}{|\mathcal{D}|} \sum_{i=1}^{|\mathcal{D}|} (x_i + \delta_i - \frac{1}{|\mathcal{D}|} \sum_{i=1}^{|\mathcal{D}|} x_i \right\|$, and assume the adversary's optimization algorithm has regret $\Theta(I^p)$ over $I$ rounds. If $\Delta \geq 2c_2 c_b I^{p-1}$, then the privacy leakage $\epsilon_p$ satisfies:*

$$\epsilon_p \leq 1 - \frac{\Delta + c_2 c_b I^{p-1}}{4D}.$$

This result suggests that by controlling $\Delta$, one can enforce an upper bound on $\epsilon_p$, thus providing a theoretical foundation for data-distortion defense mechanisms. Based on Lemma 1, we further show that the privacy-utility trade-off problem can be reformulated as a constrained data distortion problem, making it more amenable to optimization.

**Lemma 2** (Reduction to Distort-Data Problem (Zhang et al., 2024)). *Let $c = \frac{c_2 c_b I^{p-1}}{4D}$ and define $\epsilon_1 = 4D \cdot (1 - c - \epsilon)$. Then the privacy-constrained optimization:*

$$\min_{\theta} \quad \mathcal{L}(f(\theta; x + \delta), y)$$
$$s.t., \quad \epsilon_p \leq \epsilon$$

*can be reduced to:*

$$\min_{\theta} \min_{\delta} \quad \mathcal{L}(f(\theta; x + \delta), y)$$
$$s.t., \quad \|\delta\| \geq \epsilon_1$$

This reduction bridges privacy guarantees with distortion-based optimization. It enables the design of privacy-preserving mechanisms by explicitly learning data perturbations that meet privacy constraints. Moreover, by ensuring the distortion exceeds a theoretical threshold, our framework guarantees a lower bound on privacy preservation, providing formal assurance against worst-case leakage scenarios.

# D ADDITIONAL EXPERIMENTAL RESULTS

## D.1 DETAILED EXPERIMENTAL SETUP DESCRIPTION

**Privacy Metric Computation.** To quantitatively evaluate privacy leakage from gradient inversion, we employ five commonly used similarity metrics between the reconstructed image $\hat{x}$ and the original image $x$: MSE, SSIM (Wang et al., 2004), PSNR, LPIPS (Zhang et al., 2018), and KL divergence.

MSE measures the average pixel-wise squared error between two images and is computed as:

$$\text{MSE}(x, \hat{x}) = \frac{1}{n} \sum_{i=1}^{n} (x_i - \hat{x}_i)^2,$$

where $n$ is the total number of pixels.

SSIM compares two images in terms of luminance, contrast, and structure. It is computed using local image statistics:

$$\text{SSIM}(x, \hat{x}) = \frac{(2\mu_x \mu_{\hat{x}} + C_1)(2\sigma_{x\hat{x}} + C_2)}{(\mu_x^2 + \mu_{\hat{x}}^2 + C_1)(\sigma_x^2 + \sigma_{\hat{x}}^2 + C_2)},$$

where $\mu$ and $\sigma$ denote mean and standard deviation of local patches, and $C_1, C_2$ are small constants to stabilize the division.

PSNR evaluates image reconstruction quality using the MSE and is defined as:

$$\text{PSNR}(x, \hat{x}) = 10 \cdot \log_{10} \left( \frac{L^2}{\text{MSE}(x, \hat{x})} \right),$$

where $L$ is the maximum possible pixel value (e.g., 1.0 or 255 depending on normalization).

LPIPS is a learned perceptual metric that compares feature activations from a deep neural network. We use a pretrained VGG-16 model to extract features from multiple layers and computes weighted $\ell_2$ distances:

$$\text{LPIPS}(x, \hat{x}) = \sum_l \frac{1}{H_l W_l} \sum_{h,w} \| w_l \odot (\phi_l(x)_{hw} - \phi_l(\hat{x})_{hw}) \|_2^2,$$

where $\phi_l(\cdot)$ denotes the $l$-th layer's feature map, $w_l$ is a learned channel-wise weight, and $(h, w)$ indexes spatial positions.

KL divergence is used to assess semantic-level leakage by comparing the predicted label distributions of $x$ and $\hat{x}$. After passing both images through a pretrained VGG-16 classifier with softmax output, the divergence is computed as:

$$\text{KL}(P \| \hat{P}) = \sum_{i=1}^{C} P_i \log \left( \frac{P_i}{\hat{P}_i} \right),$$

where $P$ and $\hat{P}$ are the output probability distributions over $C$ classes.

**Other Settings.**  For perturbation modeling and adversarial defense, we use ResNet-18 as the default architecture. Perturbations are generated under the $L_2$ norm using PGD with random initialization enabled. For each global round, we perform 15 update steps for the perturbation model. The perturbation module is trained with a batch size of 8 with learning rate 0.1. The gradient leakage attack is implemented based on the Inverting Gradients method (Geiping et al., 2020). We optimize for 1600 steps using cosine similarity as the loss function, with a fixed learning rate of 0.1. The total variation regularization weight is set to $1 \times 10^{-5}$. Unless otherwise specified, all experiments are conducted on a single NVIDIA A6000 GPU (8 cards available).

## D.2 Convergence and Convergence Rate Analysis

Figure6 shows the time per epoch for both the SGD algorithm (without perturbation) and the perturbation algorithm (with a noise radius of 8/255) on the MNIST dataset. As shown, the time required for each epoch increases with the number of iterations $N$ needed for perturbation generation. This is expected, as the introduction of perturbations adds complexity, resulting in additional computational cost at each epoch, which is reflected in the increase in execution time.

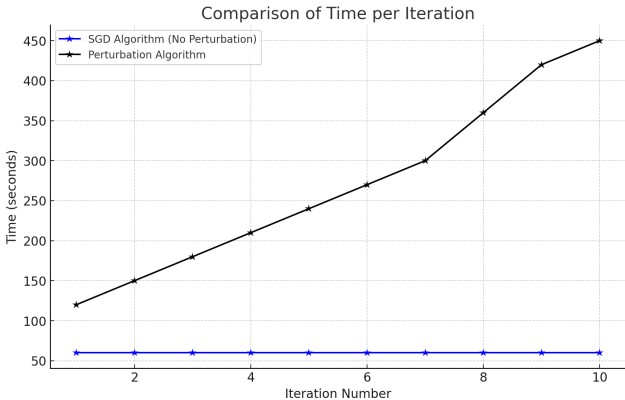

Figure 6: Time per epoch for different perturbation generation rounds on the MNIST dataset.

Figure7 demonstrates the convergence properties of our algorithm. We present the test accuracy of FedSGD (without perturbation) and FedEM with different perturbation radii on the MNIST and

FMNIST datasets. In both figures, FedEM achieves convergence within 30 training rounds, with the convergence rate closely resembling that of FedSGD. This indicates that the perturbation process does not significantly hinder the convergence speed, with both methods reaching convergence around the same number of iterations (approximately 10 rounds). These results validate that our algorithm converges efficiently even with the introduction of perturbations.

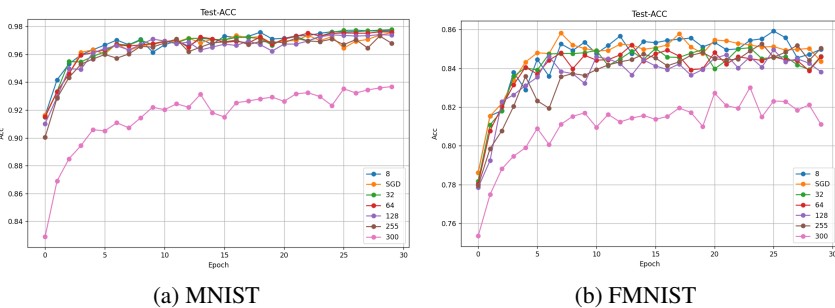

(a) MNIST
(b) FMNIST

Figure 7: Test accuracy curves for FedEM and FedSGD on the MNIST and FMNIST datasets.

### D.3 RANDOMNESS ANALYSIS

To confirm that the performance of FedEM is not an artifact of randomness, we conducted five independent runs on the MNIST dataset using different random seeds, each with a perturbation radius of $255/255$ and a gradient leakage attack launched in the first training round. Across these trials, the algorithm demonstrated strong stability in both utility and privacy metrics. The validation accuracy was $0.9747 \pm 0.0011$, and the test accuracy was $0.9750 \pm 0.0036$, showing negligible fluctuation across seeds. For privacy-related metrics, we observed similarly consistent results: MSE was $1.5373 \pm 0.0802$, PSNR was $8.4544 \pm 0.2084$, SSIM was $0.0420 \pm 0.0105$, LPIPS was $0.6618 \pm 0.0260$, and KL divergence was $3.4706 \pm 0.9623$. These results indicate that FedEM's behavior is highly reproducible and not subject to high variance under different random initializations. As further evidence, Figure 8 presents error bar plots for four representative metrics, illustrating the low variance and consistent performance of FedEM across repeated experiments.

### D.4 ADDITIONAL EXPERIMENTS OF FEDEM ON TEXT DATASETS

### D.4.1 SETTINGS

For all experiments on text classification datasets (CoLA, SST-2), we use $\text{BERT}_{\text{base}}$ as the backbone model. The LAMP-based reconstruction attack is implemented with cosine loss ($\mathcal{L}_{\cos}$) as the optimization objective, following the setup introduced in (Balunovic et al., 2022). We run the gradient inversion with $it = 30$ outer iterations, $n_c = 75$ and $n_d = 200$ inner steps, and apply early stopping once the number of total optimization steps reaches 2000. The optimizer is Adam with an initial learning rate of 1e-2, and a decay factor $\gamma$ is applied every 50 steps. To initialize the optimization, we first sample 500 embedding vectors from a standard Gaussian distribution and choose the one yielding the lowest reconstruction loss $\mathcal{L}_{\text{grad}}(x)$ as the starting point.

For defense baselines, the DP-SGD implementation uses a noise multiplier $\sigma = 0.001$ with clipping norm set to 1.0, and the Gradient Masking baseline masks 25% of randomly selected gradients during each update. FedEM uses $L_2$-bounded perturbations with radius 2.0 added in the embedding space before each local update.

### D.4.2 EXPERIMENTS ON SST2

To further evaluate the effectiveness of FedEM on textual data, we conduct experiments on the SST-2 sentiment classification dataset under the same gradient inversion attack setting. Table8 summarizes utility (MCC) and privacy leakage (ROUGE) metrics across various defense methods.

In addition to the quantitative results, we provide a representative qualitative example below. The input sentence is extracted from the SST-2 dataset. Tokens that match the original sentence are high-

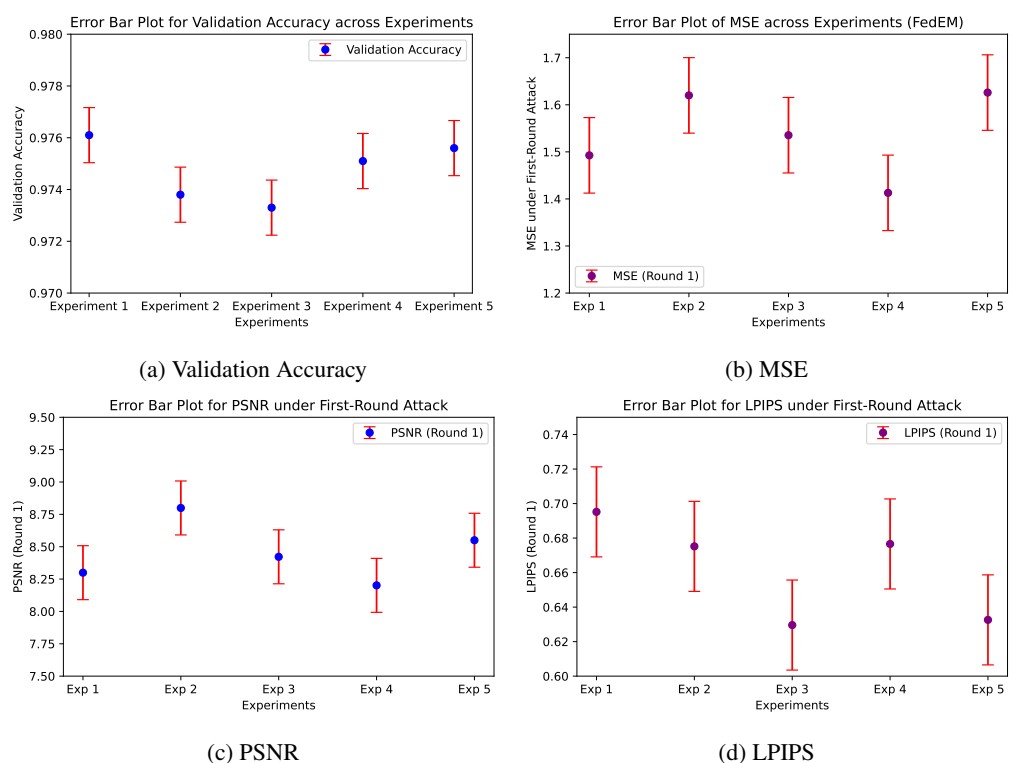

(a) Validation Accuracy      (b) MSE

(c) PSNR      (d) LPIPS

Figure 8: Error bar plots of FedEM across five random seeds on MNIST, showing stability in both utility and privacy metrics.

Table 8: Performance on the SST-2 dataset under gradient leakage attack. MCC indicates utility (↑), while ROUGE-1/2/L (%) measure the reconstruction quality of leaked text (↓). Utility metrics are marked with **U**, and privacy metrics with **P**.

| Method | MCC (U↑) | ROUGE-1 (P↓) | ROUGE-2 (P↓) | ROUGE-L (P↓) |
|---|---|---|---|---|
| FedSGD (no defend) | **0.885** | 87.7 | 74.6 | 83.8 |
| DP-SGD | 0.879 | 78.7 | 70.0 | 76.8 |
| Gradient Masked | 0.882 | 83.1 | 64.4 | 79.0 |
| FedEM (ours) | 0.882 | **78.6** | **51.0** | **73.6** |

lighted to indicate privacy leakage. Compared to baseline methods, FedEM significantly obscures key tokens, preventing accurate recovery of sensitive information.

These results further validate that FedEM effectively suppresses gradient leakage in discrete language domains, even under strong reconstruction attacks, while maintaining task performance on par with standard training.

Table 9: Reconstructed sentences under gradient leakage attacks on SST-2. Tokens matching the original input are highlighted to reflect privacy leakage. FedEM (ours) achieves the strongest protection with no direct recovery of original phrases.

| **Original** | of softheaded metaphysical claptrap |
| --- | --- |
| **FedSGD** | of soft ==metaphysical==ed of ==claptrap== |
| **DP-SGD** | ==metaphysical== cockyhort ==of== soft clapp clapped |
| **Grad-Mask** | of ==metaphysical== ==claptrap== ==softheaded== |
| **FedEM (ours)** | soft ==metaphysical== [CLS] ofhead clapped |

### D.4.3 EXPERIMENTS ON WIKITEXT-2

To further investigate the generalization ability of FedEM on discrete input tasks, we conduct additional experiments on causal language modeling with the WikiText-2 dataset. This benchmark involves discrete token-based inputs, providing a distinct evaluation scenario compared to text classification. Following the setup in (Wu et al., 2023), we adopt perplexity (PPL) as the utility metric and ROUGE-1/2/L as privacy leakage metrics.

As shown in Table 10, FedEM consistently improves privacy protection over the undefended baseline, substantially lowering ROUGE scores while keeping task utility competitive. Compared with Gaussian perturbation, FedEM achieves stronger privacy preservation (lower ROUGE-1/2/L) at a similar perplexity level. These results validate that FedEM generalizes effectively to causal language modeling, further demonstrating its robustness across both classification and generation tasks with discrete input representations.

Table 10: Causal language model training on WikiText-2 under gradient leakage attack. Perplexity indicates utility ($\downarrow$), while ROUGE-1/2/L (%) measure the reconstruction quality of leaked text ($\downarrow$). Utility metrics are marked with **U**, and privacy metrics with **P**.

| Method | ROUGE-1 (**P**$\downarrow$) | ROUGE-2 (**P**$\downarrow$) | ROUGE-L (**P**$\downarrow$) | Perplexity (**U**$\downarrow$) |
| --- | --- | --- | --- | --- |
| None (no defend) | 86.91 | 80.68 | 86.90 | **33.24** |
| Sign Compression | 64.35 | **45.40** | 64.29 | 100.32 |
| Gradient Pruning ($\alpha = 0.99$) | **64.24** | 45.79 | **64.15** | 102.56 |
| Gaussian Perturbation ($\sigma = 0.01$) | 78.75 | 67.06 | 78.71 | 50.23 |
| **FedEM (ours)** ($radius = 5$) | 68.50 | 58.00 | 68.25 | 51.12 |

## D.5 IMPACT OF PERTURBATION MAGNITUDE ON PRIVACY PROTECTION PERFORMANCE

In this section, we present all the experimental results not discussed in the main text, evaluating the performance of the proposed FedEM algorithm and comparing it against several baseline methods. The results are shown for three benchmark datasets: MNIST, FMNIST, and CIFAR-10. We analyze the privacy-utility trade-off across various perturbation magnitudes and privacy budgets.

### D.5.1 FEDEM PERFORMANCE ANALYSIS

FedEM shows a clear advantage in both privacy and utility across all datasets. On MNIST (Table11), with a perturbation radius of $8/255$, FedEM achieves the highest test accuracy (0.9767) while also providing strong privacy protection, as indicated by the low SSIM and PSNR scores. As the perturbation magnitude increases (e.g., to $32/255$), utility slightly declines, but privacy protection improves. Similar trends are observed in FMNIST (Table12) and CIFAR-10 (Table13), where FedEM consistently maintains competitive accuracy and robust privacy defense. Notably, even with

the complex CIFAR-10 dataset, FedEM outperforms other methods in terms of test accuracy while providing strong privacy metrics.

In general, we observe that moderate increases in perturbation radius improve privacy protection, but further increases lead to diminishing returns in both utility and privacy. FedEM strikes an optimal balance, achieving high privacy with minimal accuracy degradation.

Table 11: Performance of FedEM under different $L_2$-norm radius $r$ on the MNIST dataset. **E1** and **E3** denote the training round when gradient leakage attacks are launched. Utility metrics are marked with **U**, privacy metrics with **P**. Arrows indicate preferred direction: ↑ = higher is better, ↓ = lower is better.

| $r$ | Val Acc (**U**↑) | Test Acc (**U**↑) | Stage | Test MSE (**P**↑) | PSNR (**P**↓) | SSIM (**P**↓) | LPIPS (**P**↑) | KL (**P**↑) |
|---|---|---|---|---|---|---|---|---|
| 8/255 | **0.9809** | 0.9767 | E1 | 1.8251 | 7.6982 | 0.0378 | 0.6715 | 4.5235 |
| | – | – | E3 | 2.2369 | 7.0747 | 0.0445 | 0.6585 | 3.3447 |
| 32/255 | 0.9807 | **0.9777** | E1 | 1.6827 | 8.2634 | 0.0973 | 0.5768 | **4.6078** |
| | – | – | E3 | **2.2504** | **6.7443** | 0.0326 | 0.6707 | 3.3052 |
| 64/255 | 0.9803 | 0.9761 | E1 | **1.8470** | **7.5936** | **0.0346** | 0.6712 | 3.4940 |
| | – | – | E3 | 1.5830 | 8.2680 | 0.0485 | 0.6646 | 5.1168 |
| 128/255 | 0.9795 | 0.9741 | E1 | 1.7577 | 7.9013 | 0.0426 | **0.6872** | 2.9841 |
| | – | – | E3 | 2.1180 | 7.0392 | 0.0372 | 0.6183 | 5.0297 |
| 255/255 | 0.9769 | 0.9731 | E1 | 1.6208 | 8.1983 | 0.0617 | 0.6435 | 3.2251 |
| | – | – | E3 | 1.8209 | 7.7377 | **0.0266** | 0.6621 | 3.2907 |
| 300/255 | 0.9383 | 0.9368 | E1 | 1.5291 | 8.4398 | 0.0481 | 0.6550 | 3.3192 |
| | – | – | E3 | 1.4841 | 8.5576 | 0.0372 | **0.6809** | **6.5101** |

Table 12: Performance of FedEM under different $L_2$-norm radius $r$ on the FMNIST dataset. Metrics are grouped into utility (**U**) and privacy (**P**) categories. Arrows indicate desired direction: ↑ = higher is better, ↓ = lower is better.

| $r$ | Val Acc (**U**↑) | Test Acc (**U**↑) | Stage | Test MSE (**P**↑) | PSNR (**P**↓) | SSIM (**P**↓) | LPIPS (**P**↑) | KL (**P**↑) |
|---|---|---|---|---|---|---|---|---|
| 8/255 | **0.8719** | **0.8592** | E1 | 1.4988 | 7.4209 | 0.0501 | 0.6140 | 2.8601 |
| | – | – | E3 | 1.5268 | 7.2897 | 0.0650 | **0.6216** | 2.2135 |
| 32/255 | 0.8705 | 0.8506 | E1 | **1.7215** | **6.7587** | 0.0727 | 0.6077 | 2.4532 |
| | – | – | E3 | 1.5908 | 7.3026 | 0.1068 | 0.5781 | 1.3120 |
| 64/255 | 0.8664 | 0.8521 | E1 | 1.3522 | 7.8130 | 0.0755 | **0.6238** | 1.9140 |
| | – | – | E3 | 1.3803 | 7.7522 | **0.0498** | 0.6160 | 2.0367 |
| 128/255 | 0.8651 | 0.8495 | E1 | 1.1972 | 8.3873 | 0.0881 | 0.5907 | 2.0317 |
| | – | – | E3 | 1.4879 | 7.5090 | 0.0654 | 0.6052 | 2.2168 |
| 255/255 | 0.8611 | 0.8527 | E1 | 1.6642 | 6.9579 | 0.0934 | 0.5804 | 2.5536 |
| | – | – | E3 | **1.6794** | **6.8456** | 0.0527 | 0.6150 | 2.2256 |
| 300/255 | 0.8449 | 0.8301 | E1 | 1.6853 | 6.8966 | **0.0499** | 0.5422 | **3.6242** |
| | – | – | E3 | 1.4639 | 7.4637 | 0.0770 | 0.5714 | **2.3981** |

Table 13: Performance of FedEM under different $L_2$-norm radius $r$ on the CIFAR-10 dataset. Metrics are grouped into utility (**U**) and privacy (**P**) categories. Arrows indicate desired direction: ↑ = higher is better, ↓ = lower is better.

| $r$ | Val Acc (**U**↑) | Test Acc (**U**↑) | Stage | Test MSE (**P**↑) | PSNR (**P**↓) | SSIM (**P**↓) | LPIPS (**P**↑) | KL (**P**↑) |
|---|---|---|---|---|---|---|---|---|
| 8/255 | **0.2502** | **0.2518** | E1 | 2.0685 | 9.0501 | 0.0140 | **0.7954** | **3.3572** |
| | – | – | E3 | 1.6844 | 9.7758 | **0.0120** | 0.7395 | 2.2737 |
| 32/255 | 0.2468 | 0.2505 | E1 | 1.8281 | 9.4830 | 0.0151 | 0.7551 | 3.3310 |
| | – | – | E3 | 1.6826 | 9.8468 | 0.0126 | 0.7605 | 3.1947 |
| 64/255 | 0.2363 | 0.2497 | E1 | 1.7452 | 9.6302 | 0.0146 | 0.7634 | 2.0716 |
| | – | – | E3 | 1.6684 | 9.8898 | 0.0128 | 0.7574 | 3.1979 |
| 128/255 | 0.2423 | 0.2413 | E1 | **2.2336** | **8.7767** | 0.0115 | 0.7836 | 3.2235 |
| | – | – | E3 | 1.7880 | 9.6805 | 0.0123 | **0.7651** | 2.2044 |
| 255/255 | 0.2329 | 0.2331 | E1 | 1.6598 | 9.9550 | **0.0114** | 0.7448 | 2.6541 |
| | – | – | E3 | **1.9861** | **9.1988** | 0.0127 | 0.7609 | **3.8891** |
| 300/255 | 0.2261 | 0.2175 | E1 | 2.0402 | 9.0783 | 0.0131 | 0.7492 | 3.2598 |
| | – | – | E3 | 1.8592 | 9.4779 | 0.0124 | 0.7460 | 3.1268 |

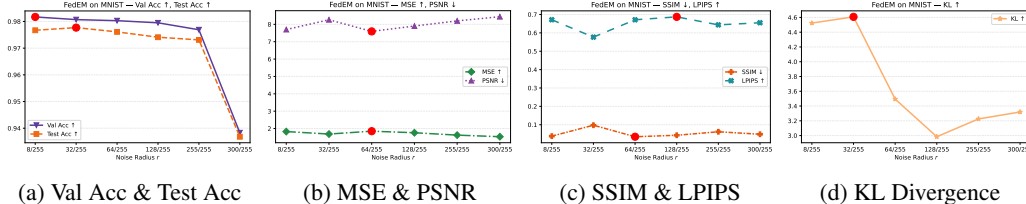

| (a) Val Acc & Test Acc | (b) MSE & PSNR | (c) SSIM & LPIPS | (d) KL Divergence |
| --- | --- | --- | --- |

Figure 9: FedEM on MNIST: Performance across 7 metrics under different $L_2$-norm radius $r$ (E1 round). Best points are highlighted in red.

### D.5.2 BASELINE COMPARISON

For comparison, we test several baseline methods: DP-Gas, DP-Lap, PPFA, and LDPM. These are evaluated under different noise scales or privacy budgets (Tables14 to 25). On MNIST, DP-based methods (Tables14 and 15) show a clear trade-off between privacy and utility. For example, DP-Gas achieves strong privacy protection with a noise scale of $16/255$ but suffers from a significant accuracy drop.

Across all datasets, while DP-based methods and LDPM offer good privacy protection at higher noise scales, they incur significant utility losses. FedEM, on the other hand, maintains high utility while still providing effective privacy protection. This demonstrates that FedEM provides a superior trade-off between privacy and utility compared to other baseline methods.

Tables 14–17 report the detailed results of four representative differential privacy mechanisms on the MNIST dataset under gradient leakage attacks. Gas-DP and Laplace-DP inject Gaussian and Laplace noise at varying scales $r$, respectively; PPFA adjusts the perturbation strength through different privacy budgets $\epsilon$; and LDPM controls noise via the standard deviation $\sigma$. Overall, these results illustrate the trade-off between privacy and utility: smaller noise (larger $\epsilon$) tends to preserve higher model accuracy but weaker privacy protection, whereas larger noise enhances resistance to reconstruction attacks at the cost of degraded utility. By comparing the four methods, we observe that Gaussian- and Laplace-based mechanisms achieve stable accuracy with gradually increasing privacy metrics, while PPFA and LDPM provide more flexible control over the privacy–utility balance.

Table 14: Performance of Gas-DP under different noise scales $r$ on the MNIST dataset. **E1** and **E3** indicate the round of federated training when the gradient leakage attack is launched (e.g., Round 1 and Round 3, respectively). Utility metrics are marked with **U**, and privacy metrics with **P**. Arrows indicate preferred direction: ↑ = higher is better, ↓ = lower is better.

| $r$ | Val Acc (**U**↑) | Test Acc (**U**↑) | Stage | Test MSE (**P**↑) | PSNR (**P**↓) | SSIM (**P**↓) | LPIPS (**P**↑) | KL (**P**↑) |
| --- | --- | --- | --- | --- | --- | --- | --- | --- |
| 1/255 | **0.9774** | **0.9741** | E1 | 1.3721 | 9.2340 | 0.1013 | 0.6321 | 3.3368 |
| | – | – | E3 | 1.5448 | 8.4097 | 0.0564 | 0.6706 | 3.2283 |
| 2/255 | 0.9759 | 0.9697 | E1 | 1.4767 | 8.6875 | 0.0401 | 0.7005 | 4.2056 |
| | – | – | E3 | 1.5465 | 8.6005 | 0.0618 | 0.6991 | 4.1331 |
| 4/255 | 0.9675 | 0.9677 | E1 | 1.5364 | 8.4700 | 0.0353 | 0.6985 | **4.2550** |
| | – | – | E3 | 1.4950 | 8.6669 | 0.0515 | 0.6992 | 2.9574 |
| 8/255 | 0.9603 | 0.9623 | E1 | **1.7068** | **7.9955** | 0.0347 | **0.7429** | 2.8834 |
| | – | – | E3 | **1.9061** | **7.4442** | 0.0253 | 0.7215 | 2.7859 |
| 16/255 | 0.9539 | 0.9537 | E1 | 1.7014 | 7.9962 | **0.0316** | 0.7391 | 3.5620 |
| | – | – | E3 | 1.8729 | 7.5252 | **0.0247** | 0.7291 | 4.5235 |

Tables 18–21 present the detailed evaluation of four representative DP mechanisms on the FMNIST dataset under gradient leakage attacks. Compared with MNIST, the overall accuracy on FMNIST is lower, reflecting the higher complexity of the dataset. Nevertheless, the same privacy–utility trade-off is observed: smaller noise or larger $\epsilon$ yields better accuracy but weaker resistance to reconstruction attacks, while larger noise enhances privacy protection at the cost of reduced model utility. Among the mechanisms, Gaussian- and Laplace-based approaches exhibit stable performance across different noise scales, while PPFA and LDPM provide flexible tuning of the balance between utility and privacy.

Table 15: Performance of Laplace-DP under different noise scales $r$ on the MNIST dataset. **E1** and **E3** indicate the round of federated training when the gradient leakage attack is launched. Utility metrics are marked with **U**, and privacy metrics with **P**. Arrows indicate preferred direction: ↑ = higher is better, ↓ = lower is better.

| $r$ | Val Acc (**U**↑) | Test Acc (**U**↑) | Stage | Test MSE (**P**↑) | PSNR (**P**↓) | SSIM (**P**↓) | LPIPS (**P**↑) | KL (**P**↑) |
|---|---|---|---|---|---|---|---|---|
| 1/255 | **0.9733** | **0.9717** | E1 | 1.3721 | 9.2340 | 0.1013 | 0.6321 | 3.3368 |
|  | – | – | E3 | 1.6264 | 8.3009 | 0.0575 | 0.6848 | 3.2296 |
| 2/255 | 0.9691 | 0.9641 | E1 | 1.5543 | 8.3723 | 0.0464 | 0.6658 | 3.2334 |
|  | – | – | E3 | 1.7452 | 7.8932 | 0.0413 | 0.7070 | 4.9510 |
| 4/255 | 0.9591 | 0.9607 | E1 | 1.5592 | 8.3509 | 0.0346 | 0.6993 | 3.1322 |
|  | – | – | E3 | **1.7557** | **7.8757** | 0.0366 | 0.7052 | 3.5539 |
| 8/255 | 0.9598 | 0.9565 | E1 | 1.8120 | 7.6918 | 0.0378 | 0.7057 | 3.1571 |
|  | – | – | E3 | 1.6472 | 8.2946 | 0.0368 | 0.7127 | 4.1443 |
| 16/255 | 0.9490 | 0.9451 | E1 | 1.7120 | 7.9328 | **0.0278** | **0.7142** | **3.5620** |
|  | – | – | E3 | 1.6930 | 7.9767 | **0.0288** | **0.7368** | **5.5580** |

Table 16: Performance of PPFA under different privacy budgets $\epsilon$ on the MNIST dataset. **E1** and **E3** indicate the round of federated training when the gradient leakage attack is launched. Utility metrics are marked with **U**, and privacy metrics with **P**. Arrows indicate preferred direction: ↑ = higher is better, ↓ = lower is better.

| $\epsilon$ | Val Acc (**U**↑) | Test Acc (**U**↑) | Stage | Test MSE (**P**↑) | PSNR (**P**↓) | SSIM (**P**↓) | LPIPS (**P**↑) | KL (**P**↑) |
|---|---|---|---|---|---|---|---|---|
| 0.995 | **0.9663** | **0.9573** | E1 | 1.2509 | 9.3820 | 0.0932 | 0.6109 | 3.5123 |
|  | – | – | E3 | 1.6359 | 8.2784 | 0.0518 | 0.6878 | 3.1741 |
| 0.99 | 0.9265 | 0.9253 | E1 | 1.2674 | 9.3565 | 0.0834 | 0.6197 | 3.4233 |
|  | – | – | E3 | 1.4712 | 8.7408 | 0.0421 | 0.7102 | 2.6431 |
| 0.98 | 0.8201 | 0.8123 | E1 | 1.2686 | 9.3460 | 0.0879 | 0.6268 | 3.4270 |
|  | – | – | E3 | 1.6543 | 8.0785 | 0.0274 | **0.7465** | 5.1111 |
| 0.97 | 0.6159 | 0.5902 | E1 | 1.3068 | 9.1965 | 0.0904 | 0.6357 | 3.4219 |
|  | – | – | E3 | **1.6528** | **8.0711** | **0.0270** | 0.7356 | **5.2627** |
| 0.8 | 0.1315 | 0.1275 | E1 | **2.0037** | **7.2492** | **0.0350** | 0.6529 | **3.6230** |
|  | – | – | E3 | 1.4323 | 8.7409 | 0.0562 | 0.7085 | 5.1851 |

Table 17: Performance of LDPM under different noise scales $\sigma$ on the MNIST dataset. **E1** and **E3** indicate the round of federated training when the gradient leakage attack is launched. Utility metrics are marked with **U**, and privacy metrics with **P**. Arrows indicate preferred direction: ↑ = higher is better, ↓ = lower is better.

| $\sigma$ | Val Acc (**U**↑) | Test Acc (**U**↑) | Stage | Test MSE (**P**↑) | PSNR (**P**↓) | SSIM (**P**↓) | LPIPS (**P**↑) | KL (**P**↑) |
|---|---|---|---|---|---|---|---|---|
| 0.0005 | **0.9756** | **0.9749** | E1 | 1.6201 | 8.2451 | 0.0527 | 0.6444 | 3.8519 |
|  | – | – | E3 | 1.6063 | 8.2799 | 0.0544 | 0.6347 | 3.0615 |
| 0.001 | 0.9733 | 0.9715 | E1 | 1.7337 | 7.8755 | 0.0598 | 0.6178 | 4.7663 |
|  | – | – | E3 | 1.6625 | 8.0371 | 0.0483 | 0.6853 | 3.0846 |
| 0.005 | 0.9727 | 0.9720 | E1 | 1.4142 | 8.8931 | 0.0440 | 0.6962 | **4.9803** |
|  | – | – | E3 | 1.8473 | 7.5874 | 0.0322 | **0.7302** | 3.7796 |
| 0.01 | 0.9605 | 0.9637 | E1 | 1.6461 | 8.1623 | 0.0445 | 0.6896 | 3.3667 |
|  | – | – | E3 | 1.8611 | 7.6997 | 0.0403 | 0.6919 | **5.1559** |
| 0.1 | 0.9103 | 0.9053 | E1 | **1.9581** | **7.3293** | **0.0294** | **0.7303** | 4.6700 |
|  | – | – | E3 | **1.8874** | **7.5364** | **0.0307** | 0.7291 | 2.9539 |

Table 18: Performance of Gas-DP under different noise scales $r$ on the FMNIST dataset. Metrics are grouped into utility (**U**) and privacy (**P**) categories. Arrows indicate the desired direction: $\uparrow$ = higher is better, $\downarrow$ = lower is better.

| $r$ | Val Acc (**U**$\uparrow$) | Test Acc (**U**$\uparrow$) | Stage | Test MSE (**P**$\uparrow$) | PSNR (**P**$\downarrow$) | SSIM (**P**$\downarrow$) | LPIPS (**P**$\uparrow$) | KL (**P**$\uparrow$) |
|---|---|---|---|---|---|---|---|---|
| 1/255 | **0.8664** | 0.8543 | E1 | 1.1693 | 8.6704 | 0.1910 | 0.5806 | 2.5319 |
| – | – | – | E3 | 1.2727 | 8.7860 | 0.1414 | 0.6067 | 2.1536 |
| 2/255 | 0.8643 | **0.8553** | E1 | 1.3268 | 7.9035 | 0.1108 | 0.6037 | 1.8837 |
| – | – | – | E3 | 1.1114 | 9.0520 | 0.1720 | 0.5728 | 2.0446 |
| 4/255 | 0.8574 | 0.8491 | E1 | 1.2918 | 8.0659 | 0.0979 | 0.5829 | 2.2742 |
| – | – | – | E3 | 1.6051 | 7.3009 | 0.0638 | 0.6619 | 2.2025 |
| 8/255 | 0.8563 | 0.8497 | E1 | 1.3753 | 7.8752 | 0.0548 | 0.6267 | **3.5238** |
| – | – | – | E3 | 1.4787 | 7.4265 | 0.0365 | 0.6634 | 1.9491 |
| 16/255 | 0.8445 | 0.8285 | E1 | **1.8619** | **6.3855** | **0.0373** | **0.6666** | 2.1791 |
| – | – | – | E3 | **1.8222** | **6.5752** | **0.0217** | **0.6677** | **3.0268** |

Table 19: Performance of Laplace-DP under different noise scales $r$ on the FMNIST dataset. Metrics are grouped into utility (**U**) and privacy (**P**) categories. Arrows indicate the desired direction: $\uparrow$ = higher is better, $\downarrow$ = lower is better.

| $r$ | Val Acc (**U**$\uparrow$) | Test Acc (**U**$\uparrow$) | Stage | Test MSE (**P**$\uparrow$) | PSNR (**P**$\downarrow$) | SSIM (**P**$\downarrow$) | LPIPS (**P**$\uparrow$) | KL (**P**$\uparrow$) |
|---|---|---|---|---|---|---|---|---|
| 1/255 | **0.8665** | 0.8497 | E1 | 1.3012 | 8.3033 | 0.1158 | 0.6052 | 1.6939 |
| – | – | – | E3 | 1.1923 | 8.6158 | 0.1381 | 0.5387 | **2.5884** |
| 2/255 | 0.8615 | **0.8535** | E1 | 1.3564 | 8.1193 | 0.1387 | 0.5923 | 1.9781 |
| – | – | – | E3 | 1.5294 | 7.3064 | 0.0568 | 0.6630 | 2.3500 |
| 4/255 | 0.8581 | 0.8488 | E1 | 1.6715 | 6.8914 | 0.0433 | 0.6496 | 3.1046 |
| – | – | – | E3 | 1.3065 | 8.0794 | 0.0628 | 0.5947 | 2.1133 |
| 8/255 | 0.8461 | 0.8361 | E1 | 1.6021 | 7.0745 | 0.0517 | 0.6283 | **3.2513** |
| – | – | – | E3 | 1.5419 | 7.4025 | 0.0410 | 0.6279 | 2.3184 |
| 16/255 | 0.8479 | 0.8331 | E1 | **1.6913** | **6.8731** | **0.0367** | **0.6759** | 1.6331 |
| – | – | – | E3 | **1.6328** | **6.9759** | **0.0272** | **0.6830** | 1.6955 |

Table 20: Performance of PPFA under different privacy budgets $\epsilon$ on the FMNIST dataset. Metrics are grouped into utility (**U**) and privacy (**P**) categories. Arrows indicate desired direction: $\uparrow$ = higher is better, $\downarrow$ = lower is better.

| $\epsilon$ | Val Acc (**U**$\uparrow$) | Test Acc (**U**$\uparrow$) | Stage | Test MSE (**P**$\uparrow$) | PSNR (**P**$\downarrow$) | SSIM (**P**$\downarrow$) | LPIPS (**P**$\uparrow$) | KL (**P**$\uparrow$) |
|---|---|---|---|---|---|---|---|---|
| 0.995 | **0.8473** | **0.8375** | E1 | 1.3615 | 8.1524 | 0.1297 | 0.5892 | 2.1581 |
| – | – | – | E3 | **1.6145** | **7.0956** | **0.0219** | **0.6581** | **2.6099** |
| 0.99 | 0.7960 | 0.7942 | E1 | 1.3522 | 8.1603 | 0.1360 | 0.5643 | 2.2359 |
| – | – | – | E3 | 1.0629 | 9.6489 | 0.6330 | 0.5743 | 2.4444 |
| 0.98 | 0.6823 | 0.6541 | E1 | 1.4535 | 7.6962 | 0.1015 | 0.5805 | 2.2431 |
| – | – | – | E3 | 1.4777 | 7.4711 | 0.0313 | 0.6357 | 2.1222 |
| 0.97 | 0.5311 | 0.5198 | E1 | 1.5789 | 7.3576 | 0.1090 | 0.5836 | 2.2918 |
| – | – | – | E3 | 1.4709 | 7.4883 | 0.0219 | 0.6534 | 2.1347 |
| 0.8 | 0.1141 | 0.1180 | E1 | **2.4718** | **5.1227** | **0.0400** | **0.6278** | **2.5256** |
| – | – | – | E3 | 1.0873 | 8.9958 | 0.0327 | 0.5994 | 1.9626 |

Table 21: Performance of LDPM under different noise scales $\sigma$ on the FMNIST dataset. Metrics are grouped into utility (**U**) and privacy (**P**) categories. Arrows indicate desired direction: $\uparrow$ = higher is better, $\downarrow$ = lower is better.

| $\sigma$ | Val Acc (**U**$\uparrow$) | Test Acc (**U**$\uparrow$) | Stage | Test MSE (**P**$\uparrow$) | PSNR (**P**$\downarrow$) | SSIM (**P**$\downarrow$) | LPIPS (**P**$\uparrow$) | KL (**P**$\uparrow$) |
|---|---|---|---|---|---|---|---|---|
| 0.0005 | **0.8715** | 0.8527 | E1 | 1.4241 | 7.8580 | 0.0877 | 0.5809 | 2.3729 |
| – | – | – | E3 | 1.5211 | 7.3467 | 0.0466 | 0.6076 | 2.0253 |
| 0.001 | 0.8693 | **0.8605** | E1 | 1.5760 | 7.4541 | 0.0959 | 0.5550 | **2.6788** |
| – | – | – | E3 | **2.0302** | **6.0581** | 0.0516 | 0.6410 | 2.0036 |
| 0.005 | 0.8653 | 0.8533 | E1 | 1.3652 | 7.9245 | 0.0625 | 0.6146 | 2.2144 |
| – | – | – | E3 | 1.5108 | 7.4389 | 0.0894 | 0.5788 | 1.6568 |
| 0.01 | 0.8610 | 0.8506 | E1 | 1.8202 | 6.5213 | 0.0583 | 0.5517 | 1.8034 |
| – | – | – | E3 | 1.7730 | 6.7465 | 0.0456 | 0.6399 | **2.6758** |
| 0.1 | 0.8043 | 0.8015 | E1 | **1.9475** | **6.2370** | **0.0314** | **0.6699** | 2.0397 |
| – | – | – | E3 | 1.8639 | 6.4218 | **0.0308** | **0.6893** | 1.6404 |

Tables 22–25 report the evaluation of four DP mechanisms on the CIFAR-10 dataset under gradient leakage attacks. Compared with MNIST and FMNIST, the overall accuracy on CIFAR-10 is substantially lower, reflecting the higher difficulty of this dataset. Nonetheless, the privacy–utility trade-off remains consistent: smaller noise or larger $\epsilon$ preserves accuracy but weakens privacy protection, whereas larger noise enhances robustness to reconstruction attacks at the expense of model utility. Among the methods, Gaussian and Laplace mechanisms show relatively stable utility as noise increases, while PPFA and LDPM provide flexible parameterization for fine-grained control over the balance between privacy and utility.

Table 22: DP-Gaussian method under varying noise scales $r$ on CIFAR10. Utility metrics are marked with **U**, and privacy metrics with **P**. Arrows indicate preferred direction: $\uparrow$ = higher is better, $\downarrow$ = lower is better.

| $r$ | Val Acc (**U**$\uparrow$) | Test Acc (**U**$\uparrow$) | Stage | Test MSE (**P**$\uparrow$) | PSNR (**P**$\downarrow$) | SSIM (**P**$\downarrow$) | LPIPS (**P**$\uparrow$) | KL (**P**$\uparrow$) |
|---|---|---|---|---|---|---|---|---|
| 1/255 | **0.2449** | 0.2504 | E1 | 1.8638 | 9.4538 | 0.0144 | 0.7549 | 2.6632 |
| – | – | – | E3 | 1.8629 | 9.4465 | 0.0131 | **0.7698** | 2.4017 |
| 2/255 | 0.2433 | **0.2581** | E1 | 1.9020 | 9.3054 | 0.0159 | 0.7351 | 3.0622 |
| – | – | – | E3 | 1.8304 | 9.4638 | 0.0165 | 0.7510 | 2.2565 |
| 4/255 | 0.2413 | 0.2381 | E1 | 1.7807 | 9.5791 | 0.0142 | 0.7606 | 2.4322 |
| – | – | – | E3 | **2.2063** | **8.7483** | 0.0159 | 0.7423 | 2.7509 |
| 8/255 | 0.2215 | 0.2159 | E1 | 2.1820 | 8.7504 | 0.0132 | 0.7615 | **4.4437** |
| – | – | – | E3 | 1.8270 | 9.4516 | **0.0125** | 0.7476 | 2.8044 |
| 16/255 | 0.2035 | 0.1973 | E1 | 2.1711 | 8.7743 | **0.0115** | **0.7649** | 4.3879 |
| – | – | – | E3 | 1.9816 | 9.1582 | 0.0140 | 0.7346 | **3.5163** |

Table 23: DP-Laplace method under varying noise scales $r$ on CIFAR10. Utility metrics are marked with **U**, and privacy metrics with **P**. Arrows indicate preferred direction: $\uparrow$ = higher is better, $\downarrow$ = lower is better.

| $r$ | Val Acc (**U**$\uparrow$) | Test Acc (**U**$\uparrow$) | Stage | Test MSE (**P**$\uparrow$) | PSNR (**P**$\downarrow$) | SSIM (**P**$\downarrow$) | LPIPS (**P**$\uparrow$) | KL (**P**$\uparrow$) |
|---|---|---|---|---|---|---|---|---|
| 1/255 | **0.2195** | **0.2213** | E1 | 1.9974 | 9.2169 | 0.0153 | 0.7601 | 2.7273 |
| – | – | – | E3 | 1.6324 | 9.9839 | 0.1117 | 0.7723 | 3.4865 |
| 2/255 | 0.2181 | 0.2210 | E1 | 1.9872 | 9.1820 | 0.0173 | 0.7576 | 4.0077 |
| – | – | – | E3 | **2.2380** | **8.8258** | 0.0145 | 0.7353 | 2.8094 |
| 4/255 | 0.2089 | 0.2123 | E1 | 1.9772 | 9.1491 | 0.0121 | 0.7657 | 2.9470 |
| – | – | – | E3 | 1.9548 | 9.1964 | 0.0145 | 0.7354 | **4.6743** |
| 8/255 | 0.1903 | 0.2045 | E1 | **2.1472** | **9.0036** | 0.0138 | **0.7903** | 4.4562 |
| – | – | – | E3 | 2.0266 | 9.1099 | **0.0114** | 0.7566 | 3.4805 |
| 16/255 | 0.1840 | 0.1817 | E1 | 2.0990 | 9.0307 | **0.0114** | 0.7841 | 3.1043 |
| – | – | – | E3 | 1.9671 | 9.1689 | 0.0135 | **0.7384** | 4.5516 |

Table 24: Performance of PPFA under varying privacy budgets $\epsilon$ on the CIFAR10 dataset. Utility metrics are marked with **U**, and privacy metrics with **P**. Arrows indicate preferred direction: $\uparrow$ = higher is better, $\downarrow$ = lower is better.

| $\epsilon$ | Val Acc (**U**$\uparrow$) | Test Acc (**U**$\uparrow$) | Stage | Test MSE (**P**$\uparrow$) | PSNR (**P**$\downarrow$) | SSIM (**P**$\downarrow$) | LPIPS (**P**$\uparrow$) | KL (**P**$\uparrow$) |
|---|---|---|---|---|---|---|---|---|
| 0.995 | 0.2489 | **0.2505** | E1 | 1.8693 | 9.4146 | 0.0152 | 0.7548 | 2.0903 |
| – | – | – | E3 | 1.8825 | 9.3554 | 0.0130 | **0.7683** | **3.3770** |
| 0.99 | **0.2527** | 0.2491 | E1 | 1.8336 | 9.4971 | **0.0152** | 0.7525 | 4.0993 |
| – | – | – | E3 | 1.7289 | 9.7083 | **0.0114** | 0.7184 | 1.8947 |
| 0.98 | 0.2437 | 0.2363 | E1 | 1.8712 | 9.4082 | 0.0180 | 0.7565 | 3.7417 |
| – | – | – | E3 | 1.6996 | 9.7810 | 0.0140 | 0.7193 | 1.9229 |
| 0.97 | 0.2393 | 0.2283 | E1 | 1.8578 | 9.4328 | 0.0172 | **0.7632** | 3.9240 |
| – | – | – | E3 | 1.8115 | 9.4606 | 0.0172 | 0.7577 | 3.3717 |
| 0.8 | 0.2047 | 0.1964 | E1 | **1.8789** | **9.3813** | 0.0162 | 0.7602 | **4.1743** |
| – | – | – | E3 | **1.9061** | **9.2433** | 0.0158 | 0.7562 | 3.5041 |

## D.6 Impact of Perturbation Lower Bounds on FedEM Performance

In this section, we provide the complete set of results related to the impact of perturbation lower bounds on FedEM's performance, which were not fully presented in the main text. These results include all the metrics evaluated at both the first (E1) and third (E3) rounds of global training, when

Table 25: LDPM performance under different noise scales $\sigma$ on the CIFAR10 dataset. Utility metrics are marked with **U**, and privacy metrics with **P**. Arrows indicate preferred direction: $\uparrow$ = higher is better, $\downarrow$ = lower is better.

| $\sigma$ | Val Acc (U$\uparrow$) | Test Acc (U$\uparrow$) | Stage | Test MSE (P$\uparrow$) | PSNR (P$\downarrow$) | SSIM (P$\downarrow$) | LPIPS (P$\uparrow$) | KL (P$\uparrow$) |
|---|---|---|---|---|---|---|---|---|
| 0.0005 | **0.2277** | **0.2278** | E1 | 2.0565 | 9.0540 | 0.0170 | 0.7455 | 3.2811 |
| | – | – | E3 | 1.9764 | 9.2572 | 0.0116 | 0.7369 | **3.4957** |
| 0.001 | 0.2135 | 0.2173 | E1 | 1.9238 | 9.3007 | 0.0139 | 0.7632 | 2.7608 |
| | – | – | E3 | 1.5831 | 10.116 | 0.0145 | **0.7575** | 3.1668 |
| 0.005 | 0.1437 | 0.1361 | E1 | **2.0639** | **9.0526** | 0.0125 | **0.7683** | 3.0425 |
| | – | – | E3 | 1.9599 | 9.1658 | 0.0109 | 0.7522 | 3.1413 |
| 0.01 | 0.1192 | 0.1191 | E1 | 2.0318 | 9.0968 | 0.0109 | 0.7518 | **4.8414** |
| | – | – | E3 | 1.6512 | 9.9284 | **0.0105** | 0.7539 | 3.1585 |
| 0.1 | 0.0938 | 0.0898 | E1 | 1.9966 | 9.0993 | **0.0101** | 0.7462 | 3.3059 |
| | – | – | E3 | **2.1834** | **8.7377** | 0.0116 | 0.7402 | 2.9869 |

gradient leakage attacks were launched. Specifically, we present utility and privacy metrics, including test and validation accuracy, MSE, SSIM, PSNR, LPIPS, and Kullback-Leibler divergence for the CIFAR-10, FMNIST, and MNIST datasets(Tables26 to 28. The tables show how varying the lower bound ($\rho_u^{\min}$) and upper bound ($\rho_u^{\max}$) on perturbation radius influences both privacy protection and model utility. These additional results further illustrate the trade-offs between privacy and accuracy under different perturbation constraints.

Table 26: Evaluation of FedEM's privacy protection under different lower bound ($\rho_u^{\min}$) and upper bound ($\rho_u^{\max}$) constraints on perturbation radius, tested on the CIFAR-10 dataset. Gradient leakage attacks are launched at epochs E1 and E3. Utility metrics are marked with **(U)** and privacy metrics with **(P)**. $\uparrow$ = higher is better, $\downarrow$ = lower is better.

| Method | Val Acc (U$\uparrow$) | Test Acc (U$\uparrow$) | Stage | MSE (P$\uparrow$) | PSNR (P$\downarrow$) | SSIM (P$\downarrow$) | LPIPS (P$\uparrow$) | KL (P$\uparrow$) |
|---|---|---|---|---|---|---|---|---|
| FedSGD (baseline) | 0.2803 | 0.2771 | E1 | 1.8219 | 9.5554 | 0.0178 | 0.7556 | 2.9228 |
| | – | – | E3 | 1.9227 | 9.3590 | 0.0131 | 0.7557 | 2.4802 |
| $\rho_u^{\max} = 200/255, \rho_u^{\min} = 25/255$ | 0.2419 | 0.2479 | E1 | 1.8512 | 9.4701 | 0.0157 | 0.7564 | 3.0267 |
| | – | – | E3 | 1.9618 | 9.1954 | 0.0129 | 0.7694 | 1.9158 |
| $\rho_u^{\max} = 200/255, \rho_u^{\min} = 50/255$ | 0.2377 | 0.2375 | E1 | 1.9897 | 9.1817 | 0.0149 | 0.7614 | 3.6679 |
| | – | – | E3 | 1.9968 | 9.2261 | 0.0128 | 0.7288 | 2.3004 |
| $\rho_u^{\max} = 200/255, \rho_u^{\min} = 100/255$ | 0.2061 | 0.2225 | E1 | 2.0333 | 9.1708 | 0.0147 | 0.7686 | 2.9418 |
| | – | – | E3 | 1.9683 | 9.2466 | 0.0147 | 0.7487 | 3.1794 |
| $\rho_u^{\max} = 400/255, \rho_u^{\min} = 100/255$ | 0.2167 | 0.2283 | E1 | 1.9006 | 9.2962 | 0.0147 | 0.7615 | 3.1473 |
| | – | – | E3 | 1.9715 | 9.2233 | 0.0148 | 0.7619 | 3.7558 |
| $\rho_u^{\max} = 400/255, \rho_u^{\min} = 100/255$ | 0.1935 | 0.1957 | E1 | 1.9620 | 9.1756 | 0.0140 | 0.7630 | 3.0510 |
| | – | – | E3 | 2.0781 | 8.9232 | 0.0111 | 0.7281 | 1.7774 |
| $\rho_u^{\max} = 400/255, \rho_u^{\min} = 200/255$ | 0.1827 | 0.2029 | E1 | 1.9163 | 9.2915 | 0.0132 | 0.7591 | 3.7197 |
| | – | – | E3 | 1.9256 | 9.3715 | 0.0140 | 0.7434 | 2.7577 |

Table 27: Evaluation of FedEM's privacy protection under different lower bound ($\rho_u^{\min}$) and upper bound ($\rho_u^{\max}$) constraints on perturbation radius, tested on the FMNIST dataset. Gradient leakage attacks are launched at epochs E1 and E3. Utility metrics are marked with **(U)** and privacy metrics with **(P)**. $\uparrow$ = higher is better, $\downarrow$ = lower is better.

| Method | Val Acc (U$\uparrow$) | Test Acc (U$\uparrow$) | Stage | MSE (P$\uparrow$) | PSNR (P$\downarrow$) | SSIM (P$\downarrow$) | LPIPS (P$\uparrow$) | KL (P$\uparrow$) |
|---|---|---|---|---|---|---|---|---|
| FedSGD (baseline) | 0.8725 | 0.8645 | E1 | 1.3711 | 8.1836 | 0.1437 | 0.5595 | 2.0664 |
| | – | – | E3 | 1.1829 | 9.4279 | 0.1741 | 0.6032 | 1.8966 |
| $\rho_u^{\max} = 200/255, \rho_u^{\min} = 25/255$ | 0.8649 | 0.8543 | E1 | 1.4090 | 7.6354 | 0.0758 | 0.6073 | 2.4429 |
| | – | – | E3 | 1.5089 | 7.3751 | 0.0402 | 0.6566 | 2.1409 |
| $\rho_u^{\max} = 200/255, \rho_u^{\min} = 50/255$ | 0.8643 | 0.8524 | E1 | 1.5766 | 7.1963 | 0.0617 | 0.6326 | 2.3450 |
| | – | – | E3 | 1.6340 | 7.0506 | 0.0540 | 0.6321 | 1.8361 |
| $\rho_u^{\max} = 200/255, \rho_u^{\min} = 100/255$ | 0.8641 | 0.8517 | E1 | 1.5972 | 7.2699 | 0.0566 | 0.6452 | 2.1789 |
| | – | – | E3 | 1.8405 | 6.5013 | 0.0453 | 0.6385 | 1.9514 |
| $\rho_u^{\max} = 400/255, \rho_u^{\min} = 50/255$ | 0.8611 | 0.8529 | E1 | 1.4470 | 7.5051 | 0.0424 | 0.6188 | 3.1473 |
| | – | – | E3 | 1.5867 | 7.0958 | 0.0789 | 0.5784 | 2.0724 |
| $\rho_u^{\max} = 400/255, \rho_u^{\min} = 100/255$ | 0.8603 | 0.8501 | E1 | 1.5906 | 7.1067 | 0.0964 | 0.5922 | 2.3557 |
| | – | – | E3 | 1.6297 | 7.1386 | 0.0517 | 0.6291 | 2.2926 |
| $\rho_u^{\max} = 400/255, \rho_u^{\min} = 200/255$ | 0.8599 | 0.8491 | E1 | 1.4463 | 7.5095 | 0.0489 | 0.6820 | 5.3285 |
| | – | – | E3 | 1.5249 | 7.2956 | 0.0667 | 0.6521 | 2.0968 |

Table 28: Evaluation of FedEM's privacy protection under different lower bound ($\rho_u^{\min}$) and upper bound ($\rho_u^{\max}$) constraints on perturbation radius, tested on the MNIST dataset. Gradient leakage attacks are launched at epochs E1 and E3. Utility metrics are marked with (U) and privacy metrics with (P). ↑ = higher is better, ↓ = lower is better.

| Method | Val Acc (U↑) | Test Acc (U↑) | Stage | MSE (P↑) | PSNR (P↓) | SSIM (P↓) | LPIPS (P↑) | KL (P↑) |
|---|---|---|---|---|---|---|---|---|
| FedSGD (baseline) | 0.9817 | 0.9753 | E1 | 1.2483 | 9.4434 | 0.1230 | 0.6192 | 2.8710 |
| | – | – | E3 | 1.3168 | 9.1517 | 0.0917 | 0.6096 | 3.3839 |
| $\rho_u^{\max} = 200/255, \rho_u^{\min} = 25/255$ | 0.9771 | 0.9759 | E1 | 1.4718 | 8.7485 | 0.0452 | 0.6561 | 3.0776 |
| | – | – | E3 | 1.7650 | 7.8324 | 0.0363 | 0.6771 | 3.7129 |
| $\rho_u^{\max} = 200/255, \rho_u^{\min} = 50/255$ | 0.9745 | 0.9735 | E1 | 1.7456 | 7.8552 | 0.0277 | 0.7169 | 4.6996 |
| | – | – | E3 | 1.7740 | 7.7798 | 0.0297 | 0.6937 | 4.5944 |
| $\rho_u^{\max} = 200/255, \rho_u^{\min} = 100/255$ | 0.9733 | 0.9695 | E1 | 1.6965 | 7.9479 | 0.0362 | 0.6715 | 3.6726 |
| | – | – | E3 | 1.5899 | 8.4807 | 0.0705 | 0.6451 | 3.4587 |
| $\rho_u^{\max} = 400/255, \rho_u^{\min} = 50/255$ | 0.9759 | 0.9749 | E1 | 1.4344 | 8.8162 | 0.0531 | 0.6589 | 5.4373 |
| | – | – | E3 | 1.7194 | 7.9133 | 0.0364 | 0.6691 | 3.3634 |
| $\rho_u^{\max} = 400/255, \rho_u^{\min} = 100/255$ | 0.9747 | 0.9723 | E1 | 1.5004 | 8.6315 | 0.0596 | 0.6283 | 3.9083 |
| | – | – | E3 | 1.8641 | 7.7520 | 0.0529 | 0.6350 | 4.4288 |
| $\rho_u^{\max} = 400/255, \rho_u^{\min} = 200/255$ | 0.9720 | 0.9729 | E1 | 1.6677 | 8.0892 | 0.0468 | 0.6878 | 3.2531 |
| | – | – | E3 | 1.8285 | 7.6805 | 0.0274 | 0.6799 | 3.8380 |

Figure 10 and 11 report the normalized test accuracy and three privacy metrics (MSE, SSIM, KL) on MNIST, FMNIST and CIFAR-10. For consistency, SSIM values are reversed during normalization so that higher values uniformly indicate stronger privacy protection.

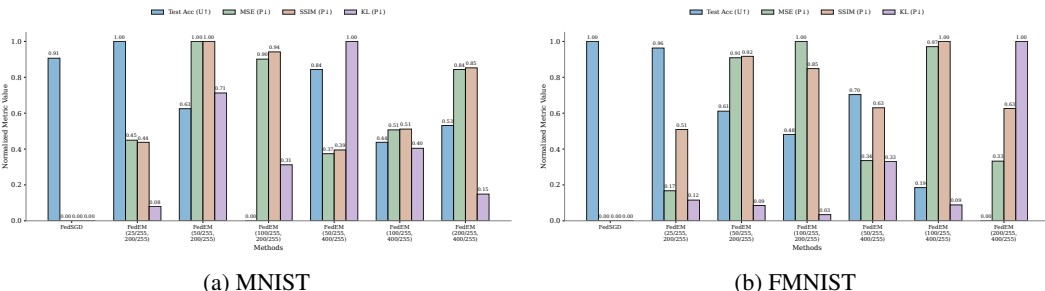

(a) MNIST                                       (b) FMNIST

Figure 10: Normalized comparison of utility and privacy metrics under different perturbation lower bounds on MNIST and FMNIST datasets. When the perturbation is constrained by a non-zero lower bound, FedEM provides a bounded privacy leakage. (The left and right endpoints of each bar denote the lower and upper bounds of the perturbation, respectively.)

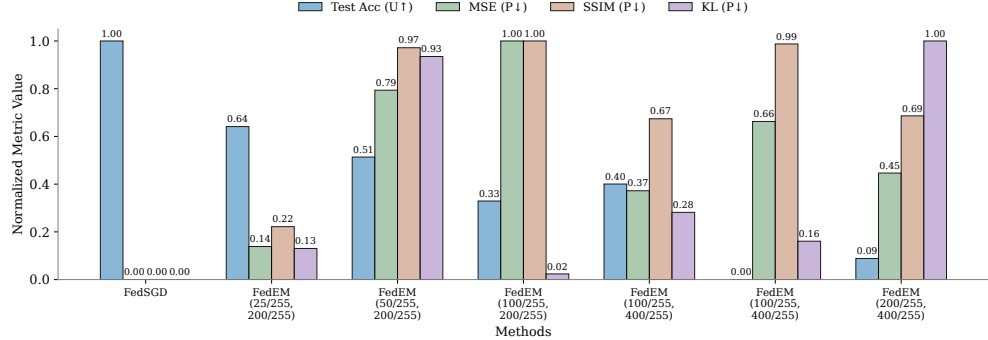

Figure 11: Normalized comparison of utility and privacy metrics under different perturbation lower bounds on CIFAR-10.

# E  PROOFS OF CONVERGENCE ANALYSIS

We provide the full assumptions, lemmas, and proof of Theorem 1.

## E.1  ASSUMPTIONS

**Assumption 1** (Smoothness). *The global objective $f(\theta)$ is $L$-smooth: $\|\nabla f(\theta) - \nabla f(\theta')\| \leq L\|\theta - \theta'\|$.*

**Assumption 2** (Bounded stochastic variance). *For any client $k$, $\mathbb{E}\big[\|g_k(\theta; x, y) - \nabla f_k(\theta)\|^2\big] \leq \sigma^2$, where $g_k(\theta; x, y) = \nabla_\theta \ell(f_\theta(x), y)$.*

**Assumption 3** (Heterogeneity). *Client dissimilarity is bounded: $\frac{1}{K} \sum_{k=1}^{K} \|\nabla f_k(\theta) - \nabla f(\theta)\|^2 \leq \zeta^2$.*

**Assumption 4** (Bounded perturbation). *Each perturbation satisfies $\|\delta_k^t\| \leq \rho_u^{max}$. Moreover, there exists $G_x > 0$ such that $\|\nabla_\theta \ell(f_\theta(x + \delta), y) - \nabla_\theta \ell(f_\theta(x), y)\| \leq G_x\|\delta\|$ for $\|\delta\| \leq \rho_u^{max}$.*

**Assumption 5** (Client sampling). *At each update, a subset $C_t$ of size $S$ is sampled uniformly, and the server aggregates $\tilde{g}^t = \frac{1}{S} \sum_{k \in C_t} \tilde{g}_k^t$. Let $\xi^t := \tilde{g}^t - \mathbb{E}[\tilde{g}^t \mid \theta^t]$ denote the sampling noise; we assume it is conditionally zero-mean, i.e., $\mathbb{E}[\xi^t \mid \theta^t] = 0$.*

## E.2  PERTURBATION BIAS LEMMA

**Lemma 3** (Bias induced by perturbation). *Let $\tilde{g}_k(\theta; x, y, \delta) = \nabla_\theta \ell(f_\theta(x + \delta), y)$ and $g_k(\theta; x, y) = \nabla_\theta \ell(f_\theta(x), y)$. Under Assumption 4,*

$$\|\mathbb{E}[\tilde{g}_k] - \nabla f_k(\theta)\| \leq G_x\rho_u^{max}, \qquad \mathbb{E}\|\tilde{g}_k - \nabla f_k(\theta)\|^2 \leq 2\sigma^2 + 2G_x^2\rho_u^{max2}.$$

*Proof.* By Lipschitz continuity,

$$\|\mathbb{E}[\tilde{g}_k] - \nabla f_k(\theta)\| = \|\mathbb{E}[\tilde{g}_k - g_k]\| \leq \mathbb{E}\|\tilde{g}_k - g_k\| \leq G_x\rho_u^{max}.$$

For the variance, observe $\|\tilde{g}_k - \nabla f_k(\theta)\|^2 \leq 2\|\tilde{g}_k - g_k\|^2 + 2\|g_k - \nabla f_k(\theta)\|^2$. Taking expectations and invoking Assumptions 2 and 4 yields the claim. $\square$

## E.3  PROOF OF THEOREM 1

*Proof.* The server update is $\theta^{t+1} = \theta^t - \eta\tilde{g}^t$ with $\tilde{g}^t = \frac{1}{S} \sum_{k \in C_t} \tilde{g}_k^t$. By $L$-smoothness of $f$ we have

$$f(\theta^{t+1}) \leq f(\theta^t) - \eta\langle\nabla f(\theta^t), \tilde{g}^t\rangle + \frac{L\eta^2}{2}\|\tilde{g}^t\|^2.$$

We decompose the aggregated update as $\tilde{g}^t = \nabla f(\theta^t) + b^t + \xi^t$, where $b^t := \mathbb{E}[\tilde{g}^t \mid \theta^t] - \nabla f(\theta^t)$ is the perturbation bias. By Lemma 3, $\|b^t\| \leq G_x\rho_u^{max}$.

Taking conditional expectation and using $\mathbb{E}[\xi^t \mid \theta^t] = 0$,

$$\mathbb{E}\big[\langle\nabla f(\theta^t), \tilde{g}^t\rangle \mid \theta^t\big] = \|\nabla f(\theta^t)\|^2 + \langle\nabla f(\theta^t), b^t\rangle \geq \tfrac{1}{2}\|\nabla f(\theta^t)\|^2 - \tfrac{1}{2}\|b^t\|^2,$$

where the last step applies Young's inequality $2\langle a, b\rangle \geq -\|a\|^2 - \|b\|^2$.

Using $\|u + v + w\|^2 \leq 3(\|u\|^2 + \|v\|^2 + \|w\|^2)$,

$$\mathbb{E}\big[\|\tilde{g}^t\|^2 \mid \theta^t\big] \leq 3\|\nabla f(\theta^t)\|^2 + 3\|b^t\|^2 + 3\mathbb{E}\big[\|\xi^t\|^2 \mid \theta^t\big].$$

Moreover, by Lemma 3 and uniform sampling of size $S$,

$$\mathbb{E}\big[\|\xi^t\|^2 \mid \theta^t\big] \leq \frac{2\sigma^2 + 2G_x^2\rho_u^{max2}}{S} + \frac{\zeta^2}{S}.$$

Combining with $\|b^t\| \leq G_x\rho_u^{max}$ gives

$$\mathbb{E}\big[\|\tilde{g}^t\|^2 \mid \theta^t\big] \leq 3\|\nabla f(\theta^t)\|^2 + 3G_x^2\rho_u^{max2} + \frac{3}{S}\big(2\sigma^2 + 2G_x^2\rho_u^{max2} + \zeta^2\big).$$

Taking expectations and substituting the two estimates,

$$\mathbb{E}[f(\theta^{t+1})] \leq \mathbb{E}[f(\theta^t)] - \eta\Big(\tfrac{1}{2}\mathbb{E}\|\nabla f(\theta^t)\|^2 - \tfrac{1}{2}\mathbb{E}\|b^t\|^2\Big)$$

$$+ \tfrac{L\eta^2}{2}\Big(3\,\mathbb{E}\|\nabla f(\theta^t)\|^2 + 3G_x^2\rho_u^{\max 2} + \tfrac{3}{S}\big(2\sigma^2 + 2G_x^2\rho_u^{\max 2} + \zeta^2\big)\Big)$$

$$\leq \mathbb{E}[f(\theta^t)] + \Big(-\tfrac{\eta}{2} + \tfrac{3L\eta^2}{2}\Big)\mathbb{E}\|\nabla f(\theta^t)\|^2 + \tfrac{\eta}{2}G_x^2\rho_u^{\max 2}$$

$$+ \tfrac{3L\eta^2}{2}\Big(G_x^2\rho_u^{\max 2} + \tfrac{2\sigma^2 + 2G_x^2\rho_u^{\max 2} + \zeta^2}{S}\Big).$$

Choose $\eta \leq \tfrac{1}{6L}$ so that $-\tfrac{\eta}{2} + \tfrac{3L\eta^2}{2} \leq -\tfrac{\eta}{4}$. Then

$$\mathbb{E}[f(\theta^{t+1})] \leq \mathbb{E}[f(\theta^t)] - \tfrac{\eta}{4}\mathbb{E}\|\nabla f(\theta^t)\|^2 + C_1\,\eta\,G_x^2\rho_u^{\max 2} + C_2\,\eta^2\Big(G_x^2\rho_u^{\max 2} + \tfrac{2\sigma^2 + 2G_x^2\rho_u^{\max 2} + \zeta^2}{S}\Big),$$

for absolute constants $C_1 = \tfrac{1}{2}$ and $C_2 = \tfrac{3L}{2}$.

Summing over $t = 0, \ldots, T-1$ and rearranging gives

$$\frac{1}{T}\sum_{t=0}^{T-1}\mathbb{E}\|\nabla f(\theta^t)\|^2 \leq \frac{4(f(\theta^0) - f^\star)}{\eta T} + \mathcal{O}\Big(\eta\,G_x^2\rho_u^{\max 2}\Big) + \mathcal{O}\Big(\eta\,\frac{2\sigma^2 + 2G_x^2\rho_u^{\max 2} + \zeta^2}{S}\Big).$$

Finally, choosing $\eta = \Theta(T^{-1/2})$ implies

$$\frac{1}{T}\sum_{t=0}^{T-1}\mathbb{E}\|\nabla f(\theta^t)\|^2 = \tilde{\mathcal{O}}(T^{-1/2}) + \mathcal{O}\Big(\rho_u^{\max 2}\Big) + \mathcal{O}\Big(\frac{\sigma^2 + G_x^2\rho_u^{\max 2} + \zeta^2}{S\sqrt{T}}\Big).$$

$\square$

