# OpenReview forum: "FedEM: A Privacy-Preserving Framework for Concurrent Utility Preservation in Federated Learning"
_ICLR.cc/2026/Conference — Submitted to ICLR 2026_

### Official Review · Reviewer_LHet · 2025-10-15

**Soundness:** 2
**Presentation:** 2
**Contribution:** 1
**Rating:** 2
**Confidence:** 4

**Summary:**

This paper proposes Federated Error Minimization (FedEM), a framework to defend against gradient leakage attacks (GLAs) in Federated Learning (FL). The core idea is to move away from traditional defenses that perturb gradients, such as those based on Differential Privacy, and instead introduce learnable perturbations directly into the clients' input data. The framework is inspired by error minimization attacks and formulates the defense as a joint optimization problem, where both the model parameters and the input perturbations are optimized to minimize the training loss. The authors conduct experiments on several image and text datasets, showing that FedEM can mitigate GLAs while maintaining higher model utility compared to several baseline methods.

**Strengths:**

- The paper proposes a defense mechanism that operates on the input data level rather than the more common gradient level.

- The paper tackles the issue of gradient leakage attacks in federated learning, which is a critical and highly relevant challenge in the field of privacy-preserving machine learning.

**Weaknesses:**

- The central idea of the paper is to use a min-min optimization to generate input perturbations. It is a direct application of the Error Minimization Attack (EMA) framework. The primary contribution seems to be repurposing this attack mechanism as a privacy defense. The technical leap here feels incremental rather than foundational. The work does not introduce a fundamentally new mechanism or insight.

- The proposed client-side training involves a loop where the perturbation vector and a temporary local model are updated for N steps for each batch. This introduces significant computational overhead compared to standard FL training. The impact on the system's feasibility is not discussed.

- In the experiments section, the authors should plot the full privacy-utility trade-off curves to show the effectiveness of the method.

- The defense is primarily evaluated against one main attack (Invert-Grad), and its robustness against a wider range of adaptive attacks is not explored.

- Table 1 should include the results of no defense mechanisms, along with the results of having defense mechanisms.

- In Table 1, the performance of FedEM is on par with the other methods, with any differences falling within a margin of 0.001 or 0.01. This suggests there is no significant distinction in their effectiveness.

**Questions:**

See comments.

**Details Of Ethics Concerns:**

N/A.

---

> ### Author Response · Authors · 2025-11-30
>
> Thank you for your valuable feedback and insightful suggestions. Below, I provide my responses to the points you raised:
>
> **Q1**: The core idea of FedEM is not a direct application of the EMA framework, but rather a response to certain limitations observed in the traditional DP methods commonly used in federated learning.  Specifically, we noticed that these methods have shortcomings that could potentially be addressed through data perturbation techniques.  In this paper, we applied a minimum-minimum optimization objective to address these issues, but we believe that many other data perturbation approaches could also be effective in achieving similar goals.
>
> Our work aims to inspire future research in this direction, rather than being limited to a simple extension of existing mechanisms. The use of the minimum-minimum optimization is just one approach to solving this problem, and we see it as a stepping stone to further exploration of privacy-preserving methods in federated learning.
>
> **Q2**: We acknowledge the increased computational overhead due to the N-step perturbation updates for each batch in the proposed client training process. To address this, we have provided  a curve illustrating the time increase due to the added computations in the appendix. This analysis shows the impact of the additional computational cost on system feasibility.
>
> **Q3**: In the paper, we have presented the effectiveness of different methods (including FedEM and baseline algorithms) in terms of varying privacy protection strengths, both in the main text and the appendix. However, we have not yet included the full privacy-utility trade-off curve. We will revise the paper to include this, adding scatter plots and curves to better demonstrate the method's effectiveness.
>
> **Q4**: This is a valid concern, but it seems more appropriate for future work. While adaptive attacks, where the attacker tailors their reconstruction based on knowledge of the perturbation mechanism, could indeed be more effective, we have not yet considered them in our current framework.  Our focus so far has been on gradient leakage attacks under the assumption of a generic adversary without prior knowledge of the privacy protection mechanism.  We believe this adaptive attack consideration can be explored as part of our future research.  In particular, it would be valuable to investigate how adaptive attacks might exploit known perturbation distributions and how these can be defended against.  In methods like DP, attackers can often reverse-engineer noise distributions, so similar considerations could be explored for FedEM. Since our perturbation mechanism does not rely on a fixed distribution, this creates a challenge for adaptive attacks, but further experimentation is needed to fully understand its implications.
>
> **Q5**: For the MNIST, FMNIST, and CIFAR-10 datasets, we have indirectly provided the results in other tables in the paper. However, it is indeed an oversight that we did not include the results without any defense mechanism in Table 1. In future versions, we will include the results of the FedSGD algorithm without any defense mechanism and compare them with the results with defense mechanisms.

---

### Official Review · Reviewer_BR9s · 2025-10-24

**Soundness:** 3
**Presentation:** 2
**Contribution:** 2
**Rating:** 4
**Confidence:** 4

**Summary:**

This paper proposes FedEM, a federated learning defense against gradient leakage attacks that perturbs client data directly rather than modifying gradients. The method jointly optimizes model parameters and learnable perturbations through dual-step optimization. Experiments on image datasets (MNIST, CIFAR-10/100, Tiny-ImageNet) and text tasks (CoLA, SST-2) show superior privacy-utility trade-offs compared to differential privacy baselines, achieving minimal accuracy loss (0.08% on MNIST) while significantly improving privacy metrics (46.2% MSE improvement, 69.3% SSIM reduction).

**Strengths:**

**1. Novel Defense Paradigm:** Input-space perturbation offers a fresh alternative to gradient-space defenses, with creative adaptation of error minimization concepts to privacy protection.

**2. Strong Empirical Results:** Consistently achieves better privacy-utility trade-offs than DP baselines across diverse datasets. Results on complex benchmarks (CIFAR-100, Tiny-ImageNet) demonstrate robustness.

**3. Comprehensive Evaluation:** Thorough experiments including multiple datasets, extensive ablations on perturbation magnitude (Tables 11-13), scalability to 50 clients, five privacy metrics, and generalization to text domains.

**4. Practical Insights:** Addresses perturbation lower bounds (Section 4.7), convergence behavior, and provides useful visualizations of perturbation evolution.

**Weaknesses:**

**1. Limited Theoretical Novelty:** Theorem 1 follows standard convergence analysis and Lemmas 1-2 are borrowed from Zhang et al. (2024). The paper provides only empirical privacy metrics without formal guarantees such as differential privacy bounds or reconstruction error guarantees.

**2. Computational Overhead Underexplored:** The method requires N=15 inner steps per batch, significantly increasing computational cost. While Figure 6 shows time increases, the paper lacks wall-clock time comparisons with baselines, memory overhead analysis, and practical guidance on selecting N for different scenarios.

**3. Incomplete Evaluation:** Privacy metrics focus primarily on the first training round with limited later-round analysis. The evaluation lacks adaptive attacks where adversaries know the defense, comparisons with gradient compression methods and unlearnable examples, statistical significance testing for most results, and realistic large-scale settings (default 4 clients).

**4. Clarity Issues:** Algorithm 1 is dense and the relationship between θ and θ_u needs clearer explanation. The claim of "stricter privacy-utility trade-off" in the abstract is not formally defined, and details on perturbation initialization and cross-round persistence are missing.

**5. Missing Analysis:** The paper provides limited characterization of what makes learned perturbations effective for privacy protection. There is no analysis of perturbation transferability across clients or systematic sensitivity analysis for key hyperparameters (αu, ρ_min, ρ_max).

**Questions:**

1. I'm concerned about adaptive attacks. What happens if an adversary knows you're using FedEM and specifically optimizes their reconstruction attack to account for the perturbations? Have you tested against such attacks?
2. The paper provides empirical privacy metrics, but can you offer any formal privacy guarantees? For instance, could you bound the reconstruction error or provide differential privacy guarantees under certain assumptions?
3. Most privacy results are from the first training round (E1). I'm curious whether privacy degrades as training progresses and the model becomes more accurate. What do the privacy metrics look like at rounds 10, 20, or 30?
4. It's not entirely clear from Algorithm 1—are the perturbations δk reinitialized at the start of each round, or do they persist and evolve across rounds? If they persist, how does this affect privacy guarantees over multiple rounds?
5. How sensitive is FedEM to the choice of hyperparameters? Specifically, how should practitioners choose N (perturbation steps), αu (perturbation learning rate), and the bounds [ρ_min, ρ_max] for their specific use case?
6. All experiments assume equal data partitioning across clients. How does FedEM perform under more realistic non-IID settings where clients have heterogeneous data distributions?
7. You mention connections to unlearnable examples, which also use data perturbations. Could you clarify the relationship and perhaps provide an empirical comparison? What are the key differences in how perturbations are learned?

---

> ### Author Response · Authors · 2025-11-30
>
> Thank you for your valuable feedback and insightful suggestions. Below, I provide my responses to the points you raised:
>
> **Q1**: This is a valid concern, but it seems more appropriate for future work.  While adaptive attacks, where the attacker tailors their reconstruction based on knowledge of the perturbation mechanism, could indeed be more effective, we have not yet considered them in our current framework.  Our focus so far has been on gradient leakage attacks under the assumption of a generic adversary without prior knowledge of the privacy protection mechanism.  We believe this adaptive attack consideration can be explored as part of our future research.  In particular, it would be valuable to investigate how adaptive attacks might exploit known perturbation distributions and how these can be defended against.  In methods like DP, attackers can often reverse-engineer noise distributions, so similar considerations could be explored for FedEM. Since our perturbation mechanism does not rely on a fixed distribution, this creates a challenge for adaptive attacks, but further experimentation is needed to fully understand its implications.
>
> **Q2**: We provide formal privacy guarantees in the appendix. Specifically, in Lemma 1, we demonstrate that when the data perturbation exceeds a certain lower bound, the privacy protection level will be no less than a specified threshold. This privacy protection level is quantified based on the reconstruction error.
>
> **Q3**: As shown in our experiments, we evaluated the performance of FedEM over all 30 global rounds on datasets like MNIST, FashionMNIST, and CIFAR-10. The results indicate that there is no clear, consistent pattern in privacy protection as the number of training iterations increases. For example, in FashionMNIST, privacy protection weakens in later rounds, while in CIFAR-10, privacy protection improves as training progresses. Therefore, it seems that the relationship between training rounds and privacy protection does not follow a universal trend. Thus, further exploration and additional experiments are needed to analyze the privacy-utility trade-off over different training rounds, and this could be an interesting avenue for future work.
>
> MNIST results:
>
> | Epoch | MSE | PSNR | SSIM | LPIPS | KL |
> | :---: | :---: | :---: | :---: | :---: | :---: |
> | E1 | 1.6137 | 8.2319 | 0.0467 | 0.6818 | 3.2219 |
> | E2 | 1.5510 | 8.3527 | 0.0324 | 0.7040 | 3.9650 |
> | E3 | 1.8097 | 7.7503 | 0.0451 | 0.6344 | 4.4362 |
> | E4 | 1.5901 | 8.2453 | 0.0372 | 0.7042 | 4.4108 |
> | E5 | 1.6768 | 8.0875 | 0.0460 | 0.7041 | 2.7963 |
> | E6 | 1.7741 | 7.7761 | 0.0315 | 0.6925 | 3.7805 |
> | E7 | 1.9257 | 7.4971 | 0.0361 | 0.6978 | 2.7890 |
> | E8 | 2.0488 | 7.1257 | 0.0263 | 0.7103 | 3.1759 |
> | E9 | 1.7905 | 7.7121 | 0.0293 | 0.6972 | 4.0423 |
> | E10 | 2.2325 | 6.7576 | 0.0219 | 0.7104 | 4.9464 |
> | E11 | 2.0174 | 7.1905 | 0.0207 | 0.7003 | 4.7912 |
> | E12 | 1.7813 | 7.7601 | 0.0312 | 0.6643 | 4.2745 |
> | E13 | 1.9086 | 7.4880 | 0.0292 | 0.6784 | 3.0190 |
> | E14 | 2.1455 | 6.9450 | 0.0196 | 0.7354 | 3.9900 |
> | E15 | 2.2327 | 6.7489 | 0.0259 | 0.7140 | 5.4296 |
> | E16 | 1.6875 | 7.9928 | 0.0314 | 0.7210 | 2.9643 |
> | E17 | 2.0458 | 7.1502 | 0.0241 | 0.7247 | 4.4487 |
> | E18 | 1.9333 | 7.4138 | 0.0210 | 0.7042 | 3.4025 |
> | E19 | 1.9288 | 7.4182 | 0.0304 | 0.7275 | 3.1418 |
> | E20 | 2.1215 | 6.9757 | 0.0207 | 0.7040 | 4.0217 |
> | E21 | 1.8364 | 7.6230 | 0.0274 | 0.7203 | 4.6662 |
> | E22 | 2.0762 | 7.0753 | 0.0220 | 0.7188 | 4.3655 |
> | E23 | 2.0822 | 7.0670 | 0.0205 | 0.7269 | 3.8911 |
> | E24 | 1.8784 | 7.5578 | 0.0258 | 0.7218 | 3.4229 |
> | E25 | 2.0307 | 7.1562 | 0.0226 | 0.7117 | 2.6416 |
> | E26 | 1.8872 | 7.4791 | 0.0252 | 0.6979 | 3.2545 |
> | E27 | 1.9558 | 7.3460 | 0.0256 | 0.6981 | 4.5123 |
> | E28 | 2.0631 | 7.1033 | 0.0253 | 0.7262 | 2.6743 |
> | E29 | 2.0622 | 7.1006 | 0.0233 | 0.6932 | 3.8763 |
>
> FMNIST results:
>
> | Epoch | MSE | PSNR | SSIM | LPIPS | KL |
> | :---: | :---: | :---: | :---: | :---: | :---: |
> | E1 | 1.9603 | 6.1532 | 0.0390 | 0.6607 | 2.5820 |
> | E2 | 1.9797 | 6.2483 | 0.0661 | 0.6195 | 1.6812 |
> | E3 | 1.5104 | 7.4275 | 0.0675 | 0.6367 | 2.3547 |
> | E4 | 1.3772 | 7.8396 | 0.0351 | 0.6780 | 1.8820 |
> | E5 | 1.5423 | 7.3752 | 0.0910 | 0.6407 | 2.0417 |
> | E6 | 1.6672 | 6.8569 | 0.0368 | 0.6671 | 2.1501 |
> | E7 | 1.8973 | 6.4343 | 0.0602 | 0.6445 | 2.5909 |
> | E8 | 1.6401 | 6.9244 | 0.0415 | 0.6686 | 2.1294 |
> | E9 | 1.8636 | 6.3872 | 0.0450 | 0.6766 | 2.2060 |
> | E10 | 1.6037 | 7.1943 | 0.0510 | 0.6701 | 3.6498 |
> | E11 | 1.5569 | 7.1859 | 0.0327 | 0.6433 | 3.1675 |
> | E12 | 1.6235 | 6.9806 | 0.0416 | 0.6605 | 2.9296 |
> | E13 | 1.6388 | 7.1163 | 0.0642 | 0.6524 | 2.4791 |
> | E14 | 1.5514 | 7.2819 | 0.0404 | 0.6514 | 2.3391 |
> | E15 | 1.8195 | 6.4697 | 0.0292 | 0.6699 | 1.8848 |
> | E16 | 1.6613 | 6.9153 | 0.0564 | 0.6613 | 2.0408 |
> | E17 | 1.7577 | 6.7232 | 0.0358 | 0.6851 | 1.5391 |
> | E18 | 1.7064 | 6.8249 | 0.0409 | 0.6294 | 3.3557 |
> | E19 | 1.5836 | 7.1761 | 0.0395 | 0.6370 | 2.1166 |

---

> ### Author Response · Authors · 2025-11-30
>
> FMNIST results:
>
> | Epoch | MSE | PSNR | SSIM | LPIPS | KL |
> | :---: | :---: | :---: | :---: | :---: | :---: |
> | E20 | 1.6262 | 6.9949 | 0.0356 | 0.6457 | 2.4555 |
> | E21 | 1.8319 | 6.5594 | 0.0391 | 0.6684 | 3.3859 |
> | E22 | 1.5129 | 7.3435 | 0.0300 | 0.6633 | 1.8242 |
> | E23 | 1.7624 | 6.6475 | 0.0277 | 0.6516 | 2.2437 |
> | E24 | 1.3515 | 7.7873 | 0.0256 | 0.6422 | 2.1777 |
> | E25 | 1.7831 | 6.5615 | 0.0393 | 0.6923 | 2.2937 |
> | E26 | 1.6424 | 6.9309 | 0.0372 | 0.6823 | 2.0199 |
> | E27 | 1.5859 | 7.1445 | 0.0330 | 0.6009 | 1.8411 |
> | E28 | 1.7121 | 6.8287 | 0.0332 | 0.6410 | 2.2496 |
> | E29 | 1.6234 | 6.9923 | 0.0365 | 0.6434 | 1.9540 |
>
> CIFAR-10 results:
> | Epoch | MSE | PSNR | SSIM | LPIPS | KL |
> | :---: | :---: | :---: | :---: | :---: | :---: |
> | E1 | 1.9945 | 9.1370 | 0.0166 | 0.7459 | 3.3951 |
> | E2 | 1.8128 | 9.4967 | 0.0147 | 0.7324 | 3.0086 |
> | E3 | 1.5388 | 10.2090 | 0.0158 | 0.7218 | 3.2136 |
> | E4 | 1.7833 | 9.6829 | 0.0120 | 0.7713 | 2.3634 |
> | E5 | 1.8048 | 9.6229 | 0.0149 | 0.7831 | 3.8458 |
> | E6 | 2.5286 | 8.1987 | 0.0127 | 0.7572 | 1.9263 |
> | E7 | 1.6942 | 9.7914 | 0.0131 | 0.7216 | 2.6571 |
> | E8 | 1.9945 | 9.1370 | 0.0166 | 0.7459 | 3.3951 |
> | E9 | 2.0622 | 8.9613 | 0.0154 | 0.7675 | 3.4237 |
> | E10 | 1.6492 | 9.9303 | 0.0137 | 0.7310 | 3.7006 |
> | E11 | 2.0971 | 8.8259 | 0.0123 | 0.7465 | 2.1509 |
> | E12 | 2.1182 | 8.9021 | 0.0120 | 0.7325 | 2.4247 |
> | E13 | 1.6940 | 9.7981 | 0.0163 | 0.7440 | 2.0659 |
> | E14 | 1.9692 | 9.2572 | 0.0168 | 0.7486 | 4.5794 |
> | E15 | 1.9389 | 9.2720 | 0.0119 | 0.7194 | 3.1861 |
> | E16 | 1.8750 | 9.3468 | 0.0118 | 0.7585 | 3.7626 |
> | E17 | 2.0070 | 9.1986 | 0.0134 | 0.7474 | 4.1677 |
> | E18 | 2.2302 | 8.7322 | 0.0166 | 0.7389 | 4.1372 |
> | E19 | 2.2670 | 8.6769 | 0.0128 | 0.7594 | 2.6142 |
> | E20 | 1.9074 | 9.2706 | 0.0155 | 0.7429 | 3.3990 |
> | E21 | 2.0073 | 9.1352 | 0.0139 | 0.7461 | 3.3644 |
> | E22 | 2.0292 | 9.2239 | 0.0134 | 0.7358 | 3.7994 |
> | E23 | 2.1293 | 8.8602 | 0.0162 | 0.7290 | 2.9654 |
> | E24 | 2.4314 | 8.8054 | 0.0159 | 0.7490 | 3.4305 |
> | E25 | 2.0581 | 8.9739 | 0.0147 | 0.7491 | 2.6948 |
> | E26 | 1.7332 | 9.6726 | 0.0143 | 0.7521 | 3.5636 |
> | E27 | 1.8530 | 9.4222 | 0.0152 | 0.7758 | 3.6938 |
> | E28 | 1.8084 | 9.5241 | 0.0163 | 0.7188 | 1.5651 |
> | E29 | 2.0130 | 9.2093 | 0.0163 | 0.7860 | 4.6388 |
>
> **Q5**: The perturbations δ_k are initialized separately for each batch before the client trains on the batch. According to Lemma 1, this initialization approach does not affect our privacy guarantee, as long as each perturbation meets a certain lower bound. Therefore, even if the perturbations are reinitialized at the start of each batch, the privacy protection remains intact, as the perturbation size is constrained by the lower bound, ensuring consistent privacy guarantees across multiple rounds of training.
> | N | VAL Acc | TEST Acc | MSE | PSNR | SSIM | LPIPS | KL |
> | :---: | :---: | :---: | :---: | :---: | :---: | :---: | :---: |
> | 10 | 0.9799 | 0.9813 | 1.4937 | 8.6553 | 0.0448 | 0.6665 | 2.1696 |
> | 15 | 0.9733 | 0.9781 | 1.6358 | 8.1536 | 0.0318 | 0.6773 | 2.1117 |
> | 20 | 0.9723 | 0.9749 | 1.5686 | 8.3991 | 0.0376 | 0.6680 | 4.6771 |
> | 25 | 0.9739 | 0.9771 | 1.3089 | 9.1971 | 0.0822 | 0.6133 | 2.9812 |
> | 30 | 0.9731 | 0.9757 | 1.5259 | 8.5778 | 0.0314 | 0.6833 | 4.6273 |
> | 35 | 0.9765 | 0.9759 | 1.4851 | 8.5657 | 0.0362 | 0.6801 | 2.5113 |
>
> **Q6**: To address the performance of FedEM in more realistic non-IID settings, I conducted experiments on the MNIST and CIFAR-10 datasets, where client data is heterogeneous, generated using a Dirichlet distribution with a heterogeneity factor of 0.5. The results show that FedEM still performs competitively compared to baseline methods. As shown in the tables below, FedEM achieved competitive validation and test accuracy, as well as strong privacy protection, despite the increased heterogeneity of the data across clients.
>
> MNIST Results-Non-IID
> | Method | VAL Acc | TEST Acc | MSE | PSNR | SSIM | LPIPS | KL |
> | :--- | :---: | :---: | :---: | :---: | :---: | :---: | :---: |
> | DP-Gas | 0.9485 | 0.9497 | 1.1101 | 10.1617 | 0.0732 | 0.6408 | 3.2949 |
> | DP-Lap | **0.9591**| **0.9619** | 1.2286 | 9.5453 | 0.0573 | 0.6882 | 3.2047 |
> | LDFM | 0.9508 | 0.9533 | 1.5657 | 8.3207 | 0.0421 | 0.6774 | 4.2014 |
> | PPFA | 0.9271 | 0.9287 | 1.7033 | 8.0637 | **0.0391** | 0.6773 | 3.7650 |
> |**FedEM(ours)**| 0.9567 | 0.9580 | **2.0306**| **7.2163** | 0.0547 | **0.7260** | **4.7188** |
>
> CIFAR-10 Results-Non-IID
> | Method | VAL Acc | TEST Acc | MSE | PSNR | SSIM | LPIPS | KL |
> | :--- | :---: | :---: | :---: | :---: | :---: | :---: | :---: |
> | DP-Gas | 0.1785 | 0.1851 | 1.9940 | 9.2291 | 0.0153 | 0.7418 | 2.5849 |
> | DP-Lap | 0.1797 | 0.1835 | 2.0168 | 9.0725 | 0.0183 | 0.7532 |  **3.7381** |
> | LDFM | 0.1770 | 0.1785 | 2.0.16 | 9.1342 |  **0.0106**| 0.7440 | 2.8429 |
> | PPFA | 0.1717 | 0.1833 | 1.4581 | 10.4125 | 0.0171 | 0.7593 | 1.8424 |
> | **FedEM(ours)**| **0.1799**  |**0.1865**  |**2.1162**  | **8.8714** | 0.0141 |  **0.7656** | 3.6609 |

---

> > ### Author Response · Authors · 2025-11-30
> >
> > **Q7**: While both our approach and methods like those for handling "unlearnable" samples share a similar optimization goal of preserving privacy through data perturbation, the underlying implementation and objectives are fundamentally different. Our primary innovation lies in using data perturbation for privacy protection, as opposed to traditional DP techniques that focus on adding noise to the model outputs or gradients.  This distinction means that our approach does not rely on the rigid assumptions of fixed noise distributions and instead focuses on optimizing privacy directly through the perturbation of the data itself. While DP has its own strengths, we believe our work can serve as a stepping stone for further exploration into more flexible, data-driven privacy-preserving methods that can be adapted to different scenarios in federated learning. In this sense, our approach opens up new possibilities for privacy protection that go beyond traditional methods.

---

### Official Review · Reviewer_2hhQ · 2025-10-29

**Soundness:** 3
**Presentation:** 4
**Contribution:** 2
**Rating:** 6
**Confidence:** 3

**Summary:**

This paper introduces Federated Error Minimization (FedEM), a defense framework for federated learning against gradient inversion attacks. The main difference that sets FedEM apart from prior defenses based on Differential Privacy (DP) is that instead of perturbing the gradients, FedEM introduces perturbations directly to each client's input. Each client learns their optimal perturbation and model in parallel, where the perturbations are restricted to have their L2 norms in a predefined interval. The authors evaluate FedEM experiments on benchmark datasets (including both text and image baed tasks) and compare its performance in terms of utility and privacy to other methods. They also provide a theoretical guarantee for FedEM convergence (to a neighborhood of stationary points) under smoothness and bounded variance assumptions. The main argument of the paper is that FedEm achieves better privacy-utility trade-off compared to prior methods, as demonstrated by the experiments.

**Strengths:**

1) The paper is very-well written and easy to comprehend. This is in part due to the main idea behind FedEM being relatively easy to convey. Still, the authors do a great job explaining what sets FedEM apart from the prior methods. The sections of the paper are organized and nicely flow from one to another.

2) The experiments in the paper, as the authors correctly claim, are pretty extensive and covers a variety of contexts to demonstrate their robustness, including different client sizes (4 vs. 50), different attacks (Inverting-Grad v. GIAS) and different parameters for FedEM and other methods (sec 4.5 & 4.7). Several question I had while reading each experiment were subsequently answered by later experiments, which is always a positive sign.

3) Despite limitations imposed by the page limit, the authors still manage to fit in important discussions that build intuition on what FedEM actually does. For example, Figure 4 and the discussion pertaining to it (end of section 4.2) is very helpful.

**Weaknesses:**

1) Despite generally outperforming the DP-based approaches in utility and privacy, the marginal contribution of FedEM in each benchmark is still relatively low. While the author's efforts in including experiments with different datasets and parameters deserve praise, the fact that FedEM achieves only a slightly better performance-privacy tradeoff than the other methods brings doubt into whether this trend would be continued in other settings. Overall, the narrative provided in the introduction presented the shortcomings of encryption- and DP- based techniques as the computational unscability and degrading model performance, but FedEM is not substantially different from the second category of methods. In this sense, the contributions of the method are relatively limited. While gradual improvements are of course important, finding at least one setting (in addition to those presented) in which FedEM overwhelmingly outperforms anything else would have strengthened the authors' point.

2) Overall, there is almost no discussion about the limitations/shortcomings of FedEM. For example, taking the gradient with respect to the perturbations can become increasingly difficulty with increasing input complexity, which may introduce complications in settings with higher-dimensional input formats that perhaps do not arise in the other methods. Further, while the method is tested against classical attacks, can there be new attacks designed specifically for FedEM? (E.g., if the adversary knows the minimum/maximum perturbation employed by the clients, can they incorporate this to their optimization objective, equation (2), accordingly?). Finally, the fact that relationship between privacy strength and noise magnitude is not strictly monotonic can give rise to complications when optimizing for the noise radius, whereas with a more straightforward relationship (as observed in other methods), finding the desired point in the accuracy-privacy trade-off can be more straightforward. While the method of course cannot defeat all such limitations, the paper should at least discuss some of them.

3) Some claims in the paper are more anecdotal and not justified by any results/citations. E.g., when talking about membership/property inference attacks: "However, with the development of defense strategies and a reassessment of their practical impact, recent research has shifted toward a more direct and severe threat: gradient leakage attacks".


Minor:
- In algorithm 1, why us theta_u part of the input?
- While potentially very helpful, Figure 1 can be a bit improved. Currently it is not super easy to read / its organization is a bit confusing
- Client heterogeneity is not defined in Theorem 1. While it is stated that all assumptions are in appendix E, it would be nice to have the formal terms that appear in the theorem defined/explained in the main body.
- $\rho_u^{max2}$ looks a bit weird in Theorem 1. Consider replacing it with $(\rho_u^{max})^2$
- Having FedSGD in Table 1 would be helpful since it is the first result being presented, the improvement in privacy / loss in utility compared to no defense is one of the first questions that come up. Similarly for Table 4.
- The axis names and numbers in Figures 2 and 5 are hard to read without zooming in.
- While noise radius is used to refer to the __max__ perturbation norm (at least according to section 3.4), it is not clear why this is the case.
- In section 4.2: "To ensure a fair comparison under high-utility settings, we set the privacy budgets or noise scales of each baseline as follows:" It is not clear why the following parameters achieve a "fair comparison".

**Questions:**

1) How does the server initialize perturbations for each client in algorithm 1? If it is just random, why are these perturbations not just initialized on the client level? Doesn't having the center send them to the client risks being intercepted and therefore can give rise to new types of attacks?

2) Since the relationship between privacy strength and noise magnitude is not strictly monotonic, how would one go about optimizing the radius for noise?

**Details Of Ethics Concerns:**

_

---

> ### Author Response · Authors · 2025-11-30
>
> Thank you for your valuable feedback and insightful suggestions. Below, I provide my responses to the points you raised:
>
> **Q1**: You are correct, this was a mistake in the pseudocode. In the actual implementation, the perturbation initialization is done by the clients themselves. If the server were responsible for initializing the perturbations, it would indeed introduce the risk of interception, potentially leading to new attack vectors. By allowing the clients to independently initialize the perturbations, we avoid this security risk, as the perturbations remain private and are not exposed to the server.
>
> **Q2**: Optimizing the noise radius is challenging because the relationship between privacy strength and noise magnitude is not strictly monotonic. While our lemma proves that perturbations only need to exceed a certain lower bound to guarantee a fixed level of privacy protection, the optimal trade-off between perturbation size and privacy strength still requires empirical exploration. Therefore, the ideal noise radius still needs to be determined through repeated experiments to find the point where privacy protection and model performance are best balanced.

---

### Official Review · Reviewer_Eosj · 2025-10-30

**Soundness:** 3
**Presentation:** 4
**Contribution:** 2
**Rating:** 4
**Confidence:** 4

**Summary:**

This paper proposes FedEM (Federated Error Minimization), a privacy-preserving federated learning framework that defends against gradient leakage attacks by injecting learnable perturbations directly into client data rather than adding noise to gradients. The method employs a dual-step optimization mechanism where each client alternates between updating a local perturbation vector delta_k and a local perturbation model theta_u over N inner steps per batch. The authors reformulate the federated optimization objective to incorporate perturbation constraints (rho_min_u leq norm delta_k leq rho_max_u), provide convergence guarantees under smoothness assumptions, and evaluate the approach on multiple image and text datasets against gradient inversion attacks. Experiments show that FedEM achieves competitive utility while providing stronger empirical resistance to reconstruction attacks compared to differential privacy baselines.

**Strengths:**

The paper introduces an input-level perturbation approach that differs from gradient-level noise injection methods, providing an interesting alternative viewpoint on privacy-utility tradeoffs in federated learning.
The evaluation spans multiple modalities (images and text), datasets (MNIST, FashionMNIST, CIFAR-10, CIFAR-100, Tiny-ImageNet, CoLA, SST-2), and attack methods (Inverting Gradients, GIAS, LAMP), demonstrating broad applicability.
The paper provides formal convergence guarantees (Theorem 1) for both convex and non-convex settings, establishing that the method reaches a neighborhood of stationary points.
The appendix contains thorough investigations of perturbation magnitude effects, lower bound constraints, scalability to 50 clients, and randomness analysis across multiple seeds.

**Weaknesses:**

W1- The paper does not provide formal (epsilon, delta)-differential privacy guarantees. Instead, it relies on Lemma 2 from Zhang et al. (2024) using a non-standard metric epsilon_p (Equation 9) that measures empirical reconstruction quality. When comparing FedEM against LDP baselines (DP-GAS, DP-LAP, LDPM) in Table 1, this creates an asymmetry: the baselines provide rigorous information-theoretic privacy guarantees while FedEM provides heuristic protection against observed attacks. The paper would benefit from either establishing formal DP guarantees or clarifying how practitioners should reason about privacy protection in the absence of worst-case guarantees.
W2- The experimental setup raises questions about fair comparison. Algorithm 1 suggests per-batch server aggregation (Line 6 iterates over batches, Line 16 aggregates inside the batch loop), but it is unclear whether baseline methods also aggregate per-batch or use standard per-epoch updates. If aggregation frequencies differ, FedEM may receive more server updates per stated "round" than baselines.
W3- Additionally, Table 1 compares methods under different privacy parameters: DP methods use noise scale 1/255, PPFA uses epsilon equals 0.995, LDPM uses sigma equals 0.0005, and FedEM uses radius 8/255. Converting these to comparable epsilon values would strengthen the evaluation.
W4- The connection between the stated optimization objective (Equation 3) and Algorithm 1 requires clarification. The objective formulates a min-min problem over theta and delta_k, but the algorithm introduces an auxiliary variable theta_u that is updated locally (Lines 9-12) yet discarded when computing the actual gradient sent to the server (Line 14 evaluates at global theta, not theta_u). Since theta_u is reset every batch (Line 8) and never communicated, its role in solving the stated objective is unclear. Additionally, Theorem 1 shows convergence to a neighborhood with error floor O(rho_max_u squared), though the practical implications of this convergence gap could be discussed more thoroughly.
W5- Several design choices would benefit from additional justification. Line 4 specifies that the server initializes perturbations delta_k, but the rationale for centralized versus local initialization is not provided. The constraint rho_min_u leq norm delta_k leq rho_max_u imposes a minimum perturbation magnitude, which differs from standard adversarial training approaches, and Table 6 indicates this affects accuracy. Finally, the computational overhead from N equals 15 inner steps per batch is acknowledged in Figure 6, but more detailed analysis (wall-clock time, FLOPs) comparing against baselines would help assess the practical trade-offs for resource-constrained federated environments.

**Questions:**

Q1. Does the server aggregate gradients per batch (as Algorithm 1 Line 16 suggests) or per epoch? Do baseline methods use the same aggregation frequency? How many total server updates occur per "round" for FedEM versus baselines?
Q2. Can the authors provide formal (epsilon, delta)-DP guarantees for FedEM? If not, how should practitioners reason about privacy protection when DP guarantees are not available? How does the privacy protection compare quantitatively to LDP methods under equivalent privacy budgets?
Q3. What are the accuracy results when all methods are constrained to the same number of server aggregations (not rounds, but actual communication events)? This would provide a fair comparison.
Q4.  Why is the local perturbation model theta_u necessary when gradients are evaluated at the global theta? What would happen if clients simply optimized delta_k directly using the global model without maintaining theta_u? Can you clarify the connection between Equation 3 and Algorithm 1 Line 14?
Q5. Why must the server initialize delta_k rather than having clients sample their own random initialization? What is the distribution used for initialization? Is this design choice necessary for correctness or is it arbitrary?
Q6. Can you provide the equivalent epsilon values (under standard DP accounting) for all methods in Table 1 to enable fair comparison? Specifically, what epsilon does FedEM with radius equals 8 divided by 255 correspond to?
Q7. What is the empirical or theoretical justification for requiring rho_min_u greater than 0? Table 6 shows accuracy degradation with higher rho_min_u - is this trade-off necessary, and under what conditions?

---

> ### Author Response · Authors · 2025-11-30
>
> Thank you for your valuable feedback and insightful suggestions. Below, I provide my responses to the questions you raised:
>
> **Q1**：All algorithms, including FedEM and the baseline methods, aggregate gradients after each batch, and the aggregation frequency is the same across all methods. Unlike FedEM, which applies perturbations directly to the data inputs at the client side, the baseline methods apply perturbations during gradient updates. Specifically, FedEM performs local perturbation optimization before computing gradients, and the server aggregates the gradients from all selected clients.  As a result, the total number of updates on the server in each "global round" is the same for both FedEM and the baseline methods, with one update per batch.
>
> **Q2**: We do not provide formal (ε, δ)-DP guarantees, as DP privacy theory requires fixed distributions (such as Laplace or Gaussian) and is not directly applicable to the gradient leakage attack scenario in federated learning. Instead, in Lemma 1, we provide a privacy guarantee that is tailored to gradient leakage attacks, establishing privacy protection metrics. The lemma demonstrates that, when a certain lower bound on the perturbation is enforced, the privacy protection level surpasses a specific threshold. This guarantee is more suitable for our setting than (ε, δ)-DP, as it offers a tighter privacy bound and is more aligned with the settings of our approach.
>
> Although the theoretical foundations of (ε, δ)-DP and our method differ, the required perturbation size can be inferred by using the same ε value. This allows for a fair comparison between our method and LDP methods.
>
> **Q3**: As with Q1, all methods (including FedEM and baseline methods) in our current setup have the same communication rounds and update frequency. The communication events and the corresponding aggregation operations are consistently aligned across all methods, allowing for a direct and equitable evaluation of performance.
>
> **Q4**: The introduction of the perturbation model θ_u is primarily to ensure a fair comparison with other methods, as those methods perform only one local iteration per batch.  By using a separate local perturbation model (θ_u), we ensure that the perturbation vector δ_k can be optimized within a single local update, just like the baseline methods.  If we were to directly optimize δ_k using the global model (θ) without maintaining θ_u, this would implicitly involve multiple rounds of model updates at the client side.  This would introduce unfairness in the comparison, as it would not align with the standard single local update used by the baseline methods.  The perturbation model θ_u allows us to maintain the fairness of the comparison by ensuring that each method follows the same iteration structure.
>
> Regarding the connection between Formula (3) and Line 14 of Algorithm 1, both describe the optimization process involving perturbations.  Formula (3) formalizes the objective of perturbation optimization, ensuring that the perturbation δ_k is updated to minimize the loss while maintaining its norm within a specified range.  In Line 14 of Algorithm 1, the gradient of the loss with respect to the global model (θ) is computed using the perturbed inputs (x_k + δ_k), and this gradient is uploaded to the server for aggregation.  The optimization in both the formula and the algorithm is aimed at achieving the same goal: optimizing the perturbation while preserving the model's utility.
>
> **Q5**: This was a mistake in the pseudocode; in the actual code implementation, the perturbation δ_k is initialized by the clients themselves. There is no fixed distribution used for the initialization. Any distribution can be used, as long as the norm of the perturbation remains within the specified lower and upper bounds. The choice of distribution is not critical to the design, as long as the perturbation's norm is properly constrained. This approach ensures that the initialization is flexible and allows for proper optimization in each client’s local model.
>
> **Q6**: The perturbation levels in Table 1 differ across methods due to the distinct implementation approaches and privacy mechanisms.  However, by calculating the required perturbation size for each method based on the same ε value, we ensure that the privacy protection strength is comparable across all methods.  While the specific perturbations added differ, the theoretical privacy guarantees, as discussed in Q2, are equivalent under the same ε.  This allows for a fair comparison of the privacy protection achieved by each method under an equivalent ε value.

---

> > ### Author Response · Authors · 2025-11-30
> >
> > **Q7**: Requiring ρ_min_u to be greater than 0 is essential to satisfy our privacy protection theory. The theory posits that when perturbations have a lower bound, the privacy protection level also has a minimum threshold. Therefore, we must impose a lower bound on the perturbation. The trade-off between increasing ρ_min_u and decreasing accuracy is reasonable, as larger perturbations inevitably lead to greater data distortion, which negatively impacts utility. This trade-off is both theoretically justified and supported by empirical results, as shown in Table 6.

---

### Comment · Area_Chair_8ab5 · 2025-11-30
**Too late responses**

Dear Authors,

Thank you for your responses. However, you post them way too late. They were expected by November 20th.

Please, see the message from PCs from November 12th:

>**We encourage you to post official comments responding to the reviews and update the paper within the next week (by November 20).**

With kind regards,

Your new AC

---

> ### Author Response · Authors · 2025-11-30
>
> Dear Area Chair,
>
> Thank you for your prompt response and for pointing out the timeline. We apologize for the delayed reply, which was indeed our oversight. However, as the discussion period has not yet ended, we kindly hope that you will still consider our responses to the reviewers' comments. We truly appreciate your understanding and the opportunity to address the feedback.
>
> Once again, thank you for your time and consideration.

---

### Meta-Review · Area_Chair_8ab5 · 2026-01-05

**Summary:**

The paper aims at addressing the issue of data leakage from the gradients exchanged during the collaboration in federated learning. As a solution to this problem, this submission proposes Federated Error Minimization (FedEM), an input-level defense framework that injects learnable perturbations into client data and jointly optimizes both the model and the perturbation generator. This solution moves away from the traditional defenses that perturb directly the gradients. The experiments are carried out on several image and text datasets, showing that FedEM can mitigate gradient leakage attacks while maintaining higher model utility compared to several baseline methods.

**Reviewer Concerns:**

AC: The responses by the authors were provided very late so there is no surprise that the reviewrs did not react.

Reviewer LHet is concerned about the novelty of this work. Indeed, it is a direct application of the Error Minimization Attack (EMA) framework. The primary contribution is to repurpose this attack mechanism as a privacy defense. The full full privacy-utility trade-off curves were not included in the revised manuscript. Table 1 should include the results of no defense mechanisms, which were not provided during the rebuttal.
The response to Reviewer LHet: "Below, I provide my responses to the points you raised:" sounds like a single author paper. This should be rephrased.

Reviewer Eosj enumerated many weaknesses and posed many questions. However, many points are not fully addressed, for example, the mistakes in pseudo-code or the fair comparison with the formal $(\varepsilon,\delta)$-DP is not fully addressed. The submission has to be updated.

The weaknesses enumerated by Reviewer 2hhQ are not addressed at all by the authors. For example, indeed, there are rather marginal contribution of FedEM in each benchmark.

Reviewer BR9s, similarly to reviewer Eosj, also indicates the issue with the limited theoretical analysis of the defense, especially the remaining question is how it can be fairly compared with the formal $(\varepsilon,\delta)$-DP defenses.

Overall, the submission requires further changes and incorporating the feedback from the reviewers. In the current state of the work, I recommend rejection.

**Reviewer Scores:**

Reviewr LHet: Score 2 / Confidence 4

Reviewer Eosj: Score 4 / Confidence 4

Reviewer 2hhQ: Score 6 / Confidence 3

Reviewer BR9s: Score 4 / Confidence 4

---

### Decision · Program_Chairs · 2026-01-26

Reject